# ConBatch-BAL: Batch Bayesian Active Learning under Budget Constraints

## Abstract

Varying annotation costs among data points and budget constraints can hinder the adoption of active learning strategies in real-world applications. This work introduces two Bayesian active learning strategies for batch acquisition under constraints (ConBatch-BAL), one based on dynamic thresholding and one following greedy acquisition. Both select samples using uncertainty metrics computed via Bayesian neural networks. The dynamic thresholding strategy redistributes the budget across the batch, while the greedy one selects the top-ranked sample at each step, limited by the remaining budget. Focusing on scenarios with costly data annotation and geospatial constraints, we also release two new real-world datasets containing geolocated aerial images of buildings, annotated with energy efficiency or typology classes. The ConBatch-BAL strategies are benchmarked against a random acquisition baseline on these datasets under various budget and cost scenarios. The results show that the developed ConBatch-BAL strategies can reduce active learning iterations and data acquisition costs in real-world settings, and even outperform the unconstrained baseline solutions.

## 1 Introduction

Bayesian active learning (BAL) is a method suitable for scenarios where data annotation is costly or time-consuming (Budd et al., 2021; Desai & Ghose, 2022; Moustapha et al., 2022). Guided by uncertainty metrics computed from a pool of candidate samples, BAL aims to improve model accuracy with fewer samples by iteratively selecting data points for annotation (Gal et al., 2017). Supported by key developments in Bayesian neural networks (MacKay, 1995; Neal, 2012), BAL has proven successful in practical scenarios, from medical applications (Kadota et al., 2024) and remote sensing (Haut et al., 2018) to natural language processing (Siddhant & Lipton, 2018).

Although Bayesian neural networks (BNNs) are based on solid mathematical principles (Neal, 2012), performing exact Bayesian inference on model parameters given the training data becomes a challenge in practical applications because the posterior distribution is usually complex and does not typically have a closed-form solution. To overcome this issue, approximate inference methods for BNNs have been proposed in the literature, from sampling approaches such as Monte Carlo Markov Chain (MCMC) (Papamarkou et al., 2022) and stochastic gradient MCMC variants (Ma et al., 2015) to variational methods like Bayes by backpropagation (MacKay, 1991) and Monte Carlo dropout (Gal & Ghahramani, 2016). While MCMC methods can infer more representative and accurate posterior distributions, variational inference methods, particularly Monte Carlo dropout, are more efficient and scalable at the expense of lower-precision uncertainty estimates. In practice, the choice of an approximate BNN is case-specific (Mohamadi & Amindavar, 2020; Wang & Yeung, 2020; Abdar et al., 2021), and depends on available computational resources and the required precision of model uncertainty estimates.

In BAL, uncertainty metrics are key for selecting data points for annotation. A pool of unlabelled candidate samples is ranked at each active learning iteration using an acquisition function. Various acquisition functions have been proposed (Gal, 2016), typically corresponding to a model uncertainty metric, such as predictive entropy (Shannon, 1948), variation ratios (Freeman, 1975), or mutual information (Gal et al., 2017). The latter relies on information-theoretic principles, distilling the model from total uncertainty by computing the mutual information of candidate sample predictions and BNN model parameters (Gal et al., 2017).

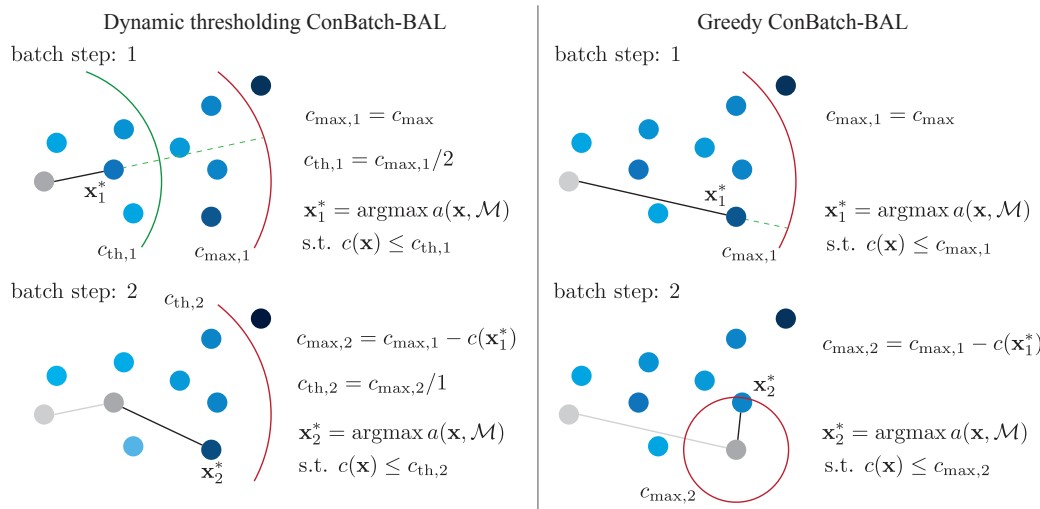

Figure 1: Proposed batch Bayesian active learning strategies, dynamic thresholding ConBatch-BAL and greedy ConBatch-BAL. (Left) Dynamic thresholding ConBatch-BAL redistributes the budget across the batch based on an adaptive threshold. (Right) Greedy ConBatch-BAL selects the top-ranked sample at each batch step, limited by the remaining budget.

BAL methods often assume a uniform annotation cost across all candidate unlabelled samples. However, in many real-world applications, annotation costs can vary, and the total budget for a batch may be limited. For example, annotating images with a drone in a geographical area is constrained by the drone's battery. To facilitate the adoption of BAL in real-world applications, we propose two active learning strategies for batch acquisition under budget constraints. Both strategies use an acquisition function that maximizes an uncertainty metric, e.g., mutual information of candidate sample predictions and model parameters. As depicted in Figure 1, adaptive thresholding ConBatch-BAL sets a cost threshold at each batch step to manage the budget, while greedy ConBatch-BAL selects the top-ranked candidate sample at each batch step, limited by the remaining budget. We also publicly release two real-world building datasets, build6k and nieman17k, where data annotation is costly and geospatial constraints are present. The datasets include geolocated building aerial images with their corresponding energy efficiency or typology classes. The proposed ConBatch-BAL strategies are tested under various budget constraints and scenarios on the building datasets and a modified version of the MNIST dataset, where each digit is geolocated.

To summarize, our contributions are listed below:

- Dynamic thresholding ConBatch-BAL and greedy ConBatch-BAL are introduced, two Bayesian active learning strategies for batch acquisition under budget constraints.
- Two curated real-world datasets are publicly released containing geolocated building aerial imagery and their corresponding building energy efficiency or typology classes.
- A benchmark study is conducted, testing the developed ConBatch-BAL strategies on real-world building datasets, and drawing insights for similar and broader practical applications.

## 2 RELATED WORK

**Batch active learning.** Batch active learning allows the annotation of multiple points simultaneously, reducing the number of model retraining iterations required. In Batch-BALD (Kirsch et al., 2019), samples are acquired based on the mutual information between a batch of points and model parameters, while its computationally efficient variant, k-BALD, computes k-wise mutual information terms (Kirsch, 2023). Other methods like BADGE (Ash et al., 2019) and ACS-FW (Pinsler et al., 2019) use loss metrics and loss gradient scores for batch selection. In Kirsch et al. (2021), a stochastic strategy is proposed for adapting single-point acquisition methods to batch active learning. Unlike our work, these methods do not consider varying data annotation costs or budget constraints.

**Cost-sensitive active learning.** Algorithms for cost-sensitive multi-class classification, where misclassification costs vary across classes, are proposed in Krishnamurthy et al. (2019), Greiner et al. (2002), and Krempl et al. (2015). In Li et al. (2022), a batch active learning algorithm is introduced for multi-fidelity physics and engineering applications, where the cost of requesting a model evaluation depends on the fidelity of the chosen model. In contrast to these approaches, our ConBatch-BAL strategies are designed for practical scenarios with varying annotation costs across data points, including cases where the acquisition cost depends on the previous set of selected points in the batch.

**Real-world aerial imagery datasets.** The AID dataset (Xia et al., 2017) contains Google Earth images for scene classification annotated by remote sensing specialists. X-View (Lam et al., 2018) and DOTA (Xia et al., 2018) are large-scale real-world datasets focused on object detection from aerial imagery, while the dataset SPAGRI-AI (Jonak et al., 2024) applies aerial imagery for benchmarking computer vision methods in crop and weed detection. In Mayer et al. (2023), a method is proposed for predicting building energy efficiency using aerial, street-view, and surface temperature data. In our work, we release two open real-world datasets containing aerial images of buildings in Rotterdam, annotated with their geolocation, energy efficiency classes, and typologies. Our goal is to motivate the advancement of active learning methods in scenarios where data annotation is costly.

## 3 PRELIMINARIES

### 3.1 PROBLEM SETTING

Consider a Bayesian neural network (BNN) model, $\mathcal{M} := \mathbf{f}^{\boldsymbol{\omega}}(\mathbf{x})$, with input $\mathbf{x}$, and model parameters $\boldsymbol{\omega} \sim p(\boldsymbol{\omega}|\mathcal{D}_{\text{train}})$, trained on a small dataset, $\mathcal{D}_{\text{train}}$. The goal of a batch-BAL task is to iteratively retrain the model, $\mathcal{M}$, based on an acquired batch of $n$ samples, $\{\mathbf{x}_1^*, \ldots, \mathbf{x}_n^*\}$, from an available pool dataset, $\mathcal{D}_{\text{pool}}$, such that the accuracy of the model on unseen data—represented by a separate test dataset, $\mathcal{D}_{\text{test}}$,— is maximized with the minimum number of iterations. The trained model provides uncertainty information on its predictions and can be used to select informative points $\{\mathbf{x}_1^*, \ldots, \mathbf{x}_n^*\} \subseteq \mathcal{D}_{\text{pool}}$ based on a given acquisition function, $a(\{\mathbf{x}_1, \ldots, \mathbf{x}_n\}, p(\boldsymbol{\omega}|\mathcal{D}_{\text{train}}))$:

$$\{\mathbf{x}_1^*, \ldots, \mathbf{x}_n^*\} = \operatorname{argmax}_{\{\mathbf{x}_{1:n}\} \subseteq \mathcal{D}_{\text{pool}}} a(\{\mathbf{x}_1, \ldots, \mathbf{x}_n\}, p(\boldsymbol{\omega}|\mathcal{D}_{\text{train}})). \quad (1)$$

Under a batch budget constraint, $c_{\max}$, allowing up to $n_{\max}$ samples per batch, and considering that the batch acquisition cost is defined as $c(\mathbf{x}_1, \ldots, \mathbf{x}_n)$, where $(\mathbf{x}_1, \ldots, \mathbf{x}_n)$ is an ordered sequence of samples, the selection process becomes:

$$\{\mathbf{x}_1^*, \ldots, \mathbf{x}_n^*\} = \operatorname*{argmax}_{\{\mathbf{x}_{1:n}\} \subseteq \mathcal{D}_{\text{pool}}} a(\{\mathbf{x}_1, \ldots, \mathbf{x}_n\}, p(\boldsymbol{\omega}|\mathcal{D}_{\text{train}})) \quad (2)$$

$$\text{subject to:} \quad c(\mathbf{x}_1, \ldots, \mathbf{x}_n) \leq c_{\max}, n \leq n_{\max}.$$

### 3.2 BATCH-BALD: BATCH BAYESIAN LEARNING BY DISAGREEMENT

To rank a batch of candidate samples, $\{\mathbf{x}_1, \ldots, \mathbf{x}_n\}$, Batch-BALD proposes an acquisition function that relies on the mutual information between sample predictions, $\{y_1, \ldots, y_n\}$, and model parameters, $p(\boldsymbol{\omega} \mid \mathcal{D}_{\text{train}})$ (Kirsch et al., 2019). In Batch-BALD, the acquisition function is defined as:

$$a_{\text{BatchBALD}}(\{\mathbf{x}_1, \ldots, \mathbf{x}_n\}, p(\boldsymbol{\omega}|\mathcal{D}_{\text{train}})) := \mathbb{I}(y_1, \ldots, y_n; \boldsymbol{\omega} \mid x_1, \ldots, x_n, \mathcal{D}_{\text{train}}). \quad (3)$$

To facilitate the selection of a diverse batch of samples, Batch-BALD does not assume independence among candidate sample predictions (Kirsch et al., 2019), and instead, the mutual information is quantified based on joint entropy metrics as:

$$\mathbb{I}(y_{1:n}; \boldsymbol{\omega} \mid \mathbf{x}_{1:n}, \mathcal{D}_{\text{train}}) := \mathbb{H}(y_{1:n} \mid \mathbf{x}_{1:n}, \mathcal{D}_{\text{train}}) - \mathbb{E}_{p(\boldsymbol{\omega}|\mathcal{D}_{\text{train}})} \mathbb{H}(y_{1:n} \mid \mathbf{x}_{1:n}, \boldsymbol{\omega}, \mathcal{D}_{\text{train}}), \quad (4)$$

where the term on the left corresponds to the joint predictive entropy, constituting a total uncertainty metric, whereas the term on the right refers to the expectation of conditional entropy given the model parameters, thereby reflecting aleatoric uncertainty. By discerning aleatoric uncertainty from joint entropy, the mutual information of a batch of samples and the model parameters becomes an *epistemic* model uncertainty metric, indicating the samples where model parameters most disagree.

### 3.3 MONTE CARLO DROPOUT

While the proposed ConBatch-BAL strategies can be generally applied to other probabilistic models, we adopt Monte Carlo dropout BNNs in this work because they scale well to high-dimensional data, are more computationally efficient than other BNN approaches, and have a straightforward training process (Kirsch et al., 2019). As demonstrated by Gal & Ghahramani (2016), minimizing the dropout loss is equivalent to performing approximate Bayesian inference, assuming the variational distribution is a mixture of two Gaussians, with a mean set equal to zero for one of them. At test time, uncertainty estimates can be obtained by computing a number of forward passes, $T$, with dropout activated. For instance, the output class probability vector for a classification task with input $\mathbf{x}$ can be estimated as $\mathbf{p}(y \mid \mathbf{x}, \mathcal{D}_{\text{train}}) \approx 1/T \sum_{t=1}^{T} \text{Softmax}(\mathbf{f}^{\boldsymbol{\omega}_t}(\mathbf{x}))$.

## 4 CONBATCH-BAL STRATEGIES

Building on methods that rely on Bayesian active learning (BAL) by disagreement (Gal et al., 2017), we introduce two heuristic strategies to handle batch acquisition under budget constraints, enabling the application of active learning methods in real-world scenarios where annotations are costly or time-consuming. Both strategies select a batch, $\{\mathbf{x}_1^*, \ldots, \mathbf{x}_n^*\}$, based on a specified uncertainty metric, such as the mutual information between predictions and model parameters, $\mathbb{I}(y_{1:n}; \boldsymbol{\omega} \mid \mathbf{x}_{1:n}, \mathcal{D}_{\text{train}})$, but differ in how they manage the remaining budget throughout the batch. The proposed strategies are named ConBatch-*BAL* because acquisition functions other than mutual information can be defined in the selection process. Sample selection under constraints is a complex combinatorial optimization problem because (i) selecting a diverse batch requires computing joint uncertainty metrics, and (ii) the acquisition cost of sample $\mathbf{x}_i$ at batch step $i$, may depend on the previous set of selected points, introducing another sequential effect to the problem. Although the definition of the cost model is case-specific, the formulation of the active learning problem under budget constraints is general. One can define the cost model and constraints in terms of monetary units, distance, or any other quantity of interest. While greedy selection based on mutual information satisfies submodularity and yields a $1 - 1/e$ approximation guarantee (Kirsch et al., 2019), introducing cost-dependent samples creates a knapsack constraint that is computationally NP-hard and may cause submodularity to no longer hold.

### 4.1 DYNAMIC THRESHOLDING CONBATCH-BAL

To manage the budget, $c_{\max}$, over a batch, dynamic thresholding initially sets a threshold $c_{th,1} = c_{\max,1}/n_{\max}$, where $n_{\max}$ is the maximum allowed number of batch steps[1] and $c_{\max,1} = c_{\max}$. At each subsequent batch step $i$, the remaining available budget is recalculated as $c_{\max,i} = c_{\max,i-1} - c(\mathbf{x}_{i-1}^*)$, based on the acquisition cost incurred by the previously selected point $\mathbf{x}_{i-1}^*$, and the batch threshold is adjusted as $c_{th,i} = c_{\max,i}/(n_{\max} - (i-1))$. Under the set budget threshold, $c_{th,i}$, a sample is selected at each batch step $i$ based on the defined acquisition function and specified uncertainty metric. The batch acquisition process[2] is summarized in Algorithm 1 and a graphical representation of two batch steps is shown in Figure 1 (left).

### 4.2 GREEDY CONBATCH-BAL

Derived from Batch-BALD (Kirsch et al., 2019), greedy ConBatch-BAL selects points based on the specified uncertainty metric, limited only by the remaining budget, operating greedily without considering the entire batch horizon. The only constraint is that the cost of the selected sample, $c(\mathbf{x}_i^*)$ must not exceed the remaining budget $c_{\max}$. This approach encourages the acquisition of costly but highly informative samples, even if it results in a batch with fewer points, $n$, than the maximum possible, $n_{\max}$. The steps for implementing this active learning strategy[2] are outlined in Algorithm 2, and the process over two batch steps is illustrated in Figure 1 (right).

---

[1]In our terminology, *batch step* refers to the stage at which a sample is selected, and *active learning iteration* denotes each instance when the model is retrained.

[2]The computational complexity per batch is primarily dominated by the computation of mutual information, $\mathcal{O}\left(|\mathcal{D}_{\text{pool}}| \cdot T \cdot K^{n_{\max}}\right)$, where $T$, $K$, and $n_{\max}$ stand for the number of forward passes, classes, and maximum points in a batch, respectively. If the joint entropy is estimated via sampling, the complexity reduces to $\mathcal{O}\left(|\mathcal{D}_{\text{pool}}| \cdot T \cdot n_{\text{sim}} \cdot K \cdot n_{\max}\right)$, with $n_{\text{sim}}$ being the number of samples used for the estimation.

---

**Algorithm 1:** Dynamic thresholding ConBatch-BAL

---

**Input:** Maximum batch acquisition size $n_{\max}$, budget $c_{\max}$, model $p(\boldsymbol{\omega} \mid \mathcal{D}_{train})$
$A_0 = \emptyset$
$c_{th} = c_{\max}/n_{\max}$
**for** $i \leftarrow 1$ *to* $n_{max}$ **do**
$\quad$ **if** $\forall \mathbf{x} \in \mathcal{D}_{pool} \setminus A_{i-1}, c(\mathbf{x}) > c_{max}$ **then**
$\quad\quad$ **return** Acquisition batch $A_{i-1} = \{\mathbf{x}_1^*, \ldots, \mathbf{x}_{i-1}^*\}$
$\quad$ **else**
$\quad\quad$ **foreach** $\mathbf{x} \in \mathcal{D}_{pool} \setminus A_{i-1} \mid c(\mathbf{x}) \leq c_{th}$ **do** $s_{\mathbf{x}} \leftarrow a(A_{i-1} \cup \{\mathbf{x}\}, p(\boldsymbol{\omega} \mid \mathcal{D}_{train}))$
$\quad\quad$ $\mathbf{x}^* \leftarrow \operatorname{argmax}_{\mathbf{x}} s_{\mathbf{x}}$
$\quad\quad$ $A_i \leftarrow A_{i-1} \cup \mathbf{x}^*$
$\quad\quad$ $c_{\max} \leftarrow c_{\max} - c(\mathbf{x}^*)$
$\quad\quad$ $c_{th} \leftarrow c_{\max}/(n_{\max} - i)$
$\quad$ **end**
**end**
**return** Acquisition batch $A_{n_{\max}} = \{\mathbf{x}_1^*, \ldots, \mathbf{x}_{n_{\max}}^*\}$

---

**Algorithm 2:** Greedy ConBatch-BAL

---

**Input:** Maximum batch acquisition size $n_{\max}$, budget $c_{\max}$, model $p(\boldsymbol{\omega} \mid \mathcal{D}_{train})$
$A_0 = \emptyset$
**for** $i \leftarrow 1$ *to* $n_{max}$ **do**
$\quad$ **if** $\forall \mathbf{x} \in \mathcal{D}_{pool} \setminus A_{i-1}, c(\mathbf{x}) > c_{max}$ **then**
$\quad\quad$ **return** Acquisition batch $A_{i-1} = \{\mathbf{x}_1^*, \ldots, \mathbf{x}_{i-1}^*\}$
$\quad$ **else**
$\quad\quad$ **foreach** $\mathbf{x} \in \mathcal{D}_{pool} \setminus A_{i-1} \mid c(\mathbf{x}) \leq c_{max}$ **do** $s_{\mathbf{x}} \leftarrow a(A_{i-1} \cup \{\mathbf{x}\}, p(\boldsymbol{\omega} \mid \mathcal{D}_{train}))$
$\quad\quad$ $\mathbf{x}^* \leftarrow \operatorname{argmax}_{\mathbf{x}} s_{\mathbf{x}}$
$\quad\quad$ $A_i \leftarrow A_{i-1} \cup \mathbf{x}^*$
$\quad\quad$ $c_{\max} \leftarrow c_{\max} - c(\mathbf{x}^*)$
$\quad$ **end**
**end**
**return** Acquisition batch $A_{n_{\max}} = \{\mathbf{x}_1^*, \ldots, \mathbf{x}_{n_{\max}}^*\}$

---

## 5 BUILDING DATASETS

BAL allows the acquisition of multiple samples simultaneously within an active learning iteration. In practical applications, this acquisition cost may vary among samples and batch selection may be constrained by a budget. For example, a drone annotator operates under a finite battery range, or a medical expert may need to travel between labs across various regions. Focusing on practical applications where data annotation is typically geospatially constrained and costly, we introduce two real-world datasets, *build6k* and *nieman17k*. These datasets consist of building aerial imagery and their corresponding building energy labels or typologies, which are key factors for assessing energy efficiency and intervention needs within a city. A few sample images from each dataset are shown in Figure 2a. The *build6k* dataset includes approximately 6,000 aerial images of buildings in Rotterdam, each classified as {efficient (energy labels A-E), inefficient (energy labels F-G)}. Categorizing building energy efficiency at scale supports recent energy performance directives aimed at progressively phasing out inefficient buildings (Economidou et al., 2020).

We also release the *nieman17k* dataset, containing approximately 17,000 building aerial images with their typology class: {upstairs apartment ($<$1945), terraced house ($<$1945), terraced house ($>$1945), porch house ($<$1945), porch house (1945-1975), porch house ($>$1975), detached house}. Building typologies can be used in energy performance simulations (Koezjakov et al., 2018), where thermal properties can be inferred as a function of building typology. Besides energy performance, typologies can also be used for architectural analyses focusing on building historical evolution.

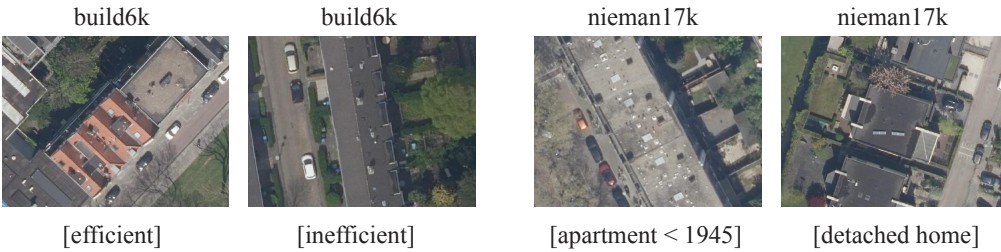

build6k      build6k      nieman17k      nieman17k

[efficient]     [inefficient]     [apartment < 1945]     [detached home]

(a) Sample aerial images from the build6k and nieman17k datasets with their corresponding class categories.

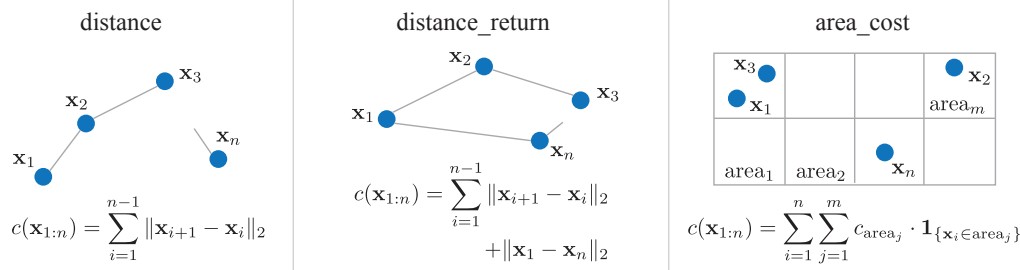

distance

$$c(\mathbf{x}_{1:n}) = \sum_{i=1}^{n-1} \|\mathbf{x}_{i+1} - \mathbf{x}_i\|_2$$

distance_return

$$c(\mathbf{x}_{1:n}) = \sum_{i=1}^{n-1} \|\mathbf{x}_{i+1} - \mathbf{x}_i\|_2 + \|\mathbf{x}_1 - \mathbf{x}_n\|_2$$

area_cost

$$c(\mathbf{x}_{1:n}) = \sum_{i=1}^{n} \sum_{j=1}^{m} c_{\mathrm{area}_j} \cdot \mathbf{1}_{\{\mathbf{x}_i \in \mathrm{area}_j\}}$$

(b) Cost configurations. (Left) The total travel distance from the first to the last point is constrained within a batch. (Center) The constrained total travel distance includes the return to the first point in the batch. (Right) Sample acquisition costs are discretized into areas and their sum is constrained within a batch.

Figure 2: Released building datasets and implemented cost configurations.

All buildings in both datasets are geo-referenced. Finally, we also include a modified version of MNIST for comparative purposes, where approximately 6,000 images are randomly linked to geolocations from build6k. Although we apply the proposed active learning strategies to scenarios with geospatial constraints, we encourage the development of real-world datasets that employ other cost models based on factors such as monetary value or time.

The aerial images in *build6k* and *nieman17k* are collected via PDOK's web service (PDOK, 2024). Each aerial image captures a building, delimited by its footprint, with geographical coordinates, energy labels, and typologies, all retrieved from open data in collaboration with the municipality of Rotterdam. In addition to aerial imagery, energy efficiency classes, and typologies, the datasets[3] include predefined training, test, and pool sets, as well as the feature vectors for each building, computed via DINOv2 (ViT-S/14 distilled)(Oquab et al., 2023). Further details on the process followed to curate the datasets are provided in Appendix A. Although this work focuses on energy efficiency and typology classification, the datasets can be used for other tasks such as information retrieval or instance segmentation (Sun et al., 2022).

Hiring auditors to inspect or classify all buildings in a city is prohibitively expensive. Carefully selecting buildings for inspection can significantly reduce costs. To motivate our experiments, we introduce three cost configurations, shown in Figure 2b, adaptable to other similar real-world applications. The first configuration, *distance*, allows free choice of the first building in a batch but constrains the distance traveled when selecting subsequent buildings in the batch, reflecting transportation costs that can be reduced by inspecting nearby buildings together. The second configuration, *distance_return*, imposes a stricter constraint by requiring the total traveled distance to include the return trip to the first building in the batch, accounting for the extra resources needed to complete the route. Finally, the configuration *area_cost* assigns each building a cost based on its geographical area in Rotterdam, with costs proportional to the number of buildings in the area, ranging from 1 to 100 cost units. Crowded areas in this configuration are more expensive than sparsely populated ones. We benchmark the proposed ConBatch-BAL strategies on the three cost configurations, presenting results for *distance* and *area_cost* in the main text, with results for *distance_return* provided in Appendix C.1, as they show minor variations compared to the *distance* configuration.

---

[3]The datasets, *build6k* and *nieman17k*, are publicly released under a CC by 4.0 license. Anonymous link.

## 6 EXPERIMENTS

In this section, we benchmark the proposed ConBatch-BAL strategies against a random selection baseline. The strategies are tested on three datasets, build6k, mnist6k, and nieman17k under various budget constraints. For each dataset, configuration, budget constraint, and strategy, 5 random seeds are evaluated over 800 active learning iterations. We evaluate the number of active learning iterations required to achieve a specified model accuracy on the test set, as this is equivalent to counting the number of completed tours in the tested practical scenarios. Additionally, we showcase the results as a function of acquired samples and cost in Appendix C.3. Note that in the limit of an infinite budget, dynamic thresholding ConBatch-BAL, greedy ConBatch-BAL, and greedy Batch-BALD become equivalent, as all acquire samples based on mutual information.

In all experiments, the Bayesian neural network classifier receives embedding vectors generated from the input aerial images using the pre-trained DINOv2 foundation model (Oquab et al., 2023). Further details are provided in Appendix B.1. A preliminary analysis showed that fine-tuning the last two layers of DINOv2 improves accuracy only by 1-2%, agreeing with reported experiments (Oquab et al., 2023). We thus restrict model re-training to the Bayesian neural network classifier, which is modeled with two hidden layers, and hyperparameters listed in Table 2 in Appendix B.2. In this setup, the specified acquisition functions correspond to the mutual information of the BNN classifier parameters and sample output predictions.

The results[4] for the *distance* cost configuration are showcased in Figure 3. Markers represent the number of active learning iterations a strategy requires to reach a specific classification accuracy for each seed. In the figure, budget constraints are color-coded. Consistent with previously reported findings (Kirsch et al., 2019), we observe that greedy Batch-BALD outperforms random selection under an infinite budget, requiring significantly fewer samples to achieve a high classification accuracy.

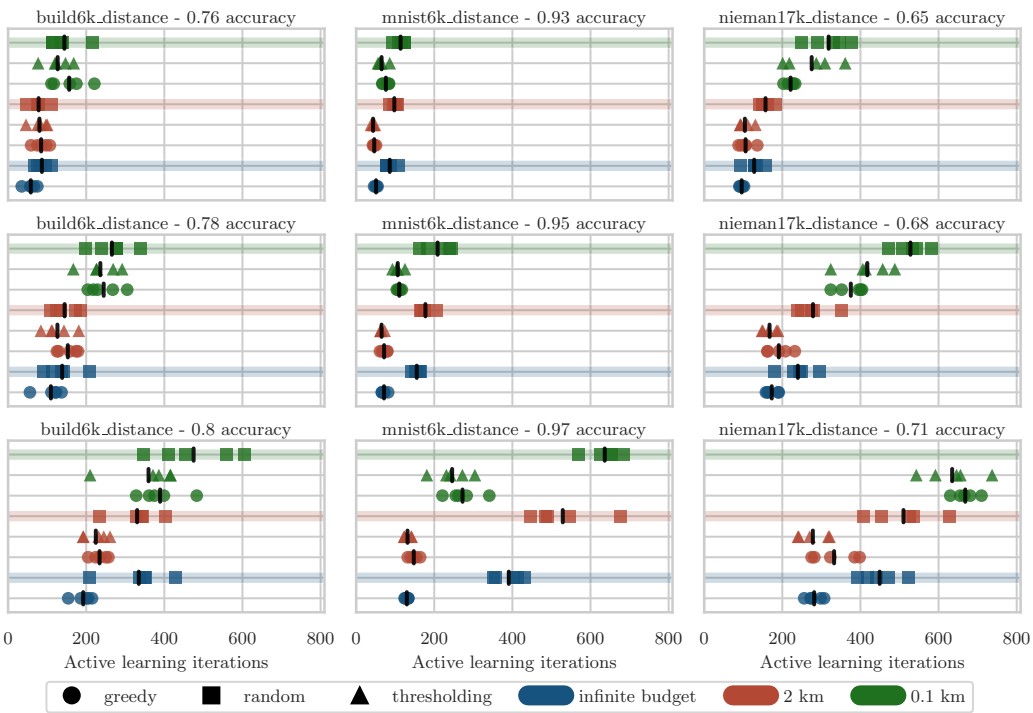

Figure 3: Benchmark results for the *distance* cost configuration across datasets and budget constraints. Markers represent the number of active learning iterations required to reach model accuracy targets on the test set for each seed, with the mean indicated by a vertical line. *Greedy ConBatch-BAL* and *dynamic thresholding ConBatch-BAL* outperform the random selection baseline.

---

[4]The corresponding active learning curves are shown in Figure 11 in Appendix C.2

In the *distance* configuration under budget constraints, dynamic thresholding ConBatch-BAL and greedy ConBatch-BAL achieve the highest model accuracy targets with 20–43% fewer active learning iterations on the build6k dataset, and 50–80% fewer on the nieman17k dataset, compared to the random selection baseline. Under the 100-m batch constraint, the baseline fails to reach the target of 71% model accuracy on the nieman17k dataset within 800 active learning iterations. With fewer than 400 active iterations, both ConBatch-BAL strategies achieve an accuracy of 97% on mnist6k under a 100-m budget constraint, while the random strategy requires over 600 active iterations to reach the same accuracy. Naturally, more active learning iterations are needed in settings with stricter budget constraints, as opportunities to select highly informative samples become more limited.

Among the datasets, the performance gap between the proposed ConBatch-BAL strategies and random selection is larger in mnist6k compared to build6k and nieman17k. We attribute this result to the fact that build6k and nieman17k are noisier, making it more difficult to distinguish model uncertainty from aleatoric noise. Nevertheless, the observed trends hold across datasets, with ConBatch-BAL strategies outperforming random selection, especially for higher classification accuracy targets. Interestingly, the developed ConBatch-BAL strategies under the 2 km constraint outperform the unconstrained random selection baseline on all datasets. As detailed in Appendix C.5, the effectiveness of ConBatch-BAL strategies relative to the baseline remains consistent across varying batch sizes.

To further investigate the proposed ConBatch-BAL strategies, we conduct additional experiments in the *cost_area* configuration, where acquisition cost depends on the area where the sample is located. In this case, the cost model is not sequential, as the sequence in which samples are acquired does not influence batch acquisition cost. While adaptive thresholding balances the available budget across the batch, highly informative yet costly samples may be overlooked in early batch steps due to the imposed thresholds, potentially remaining unreachable throughout the entire batch selection process. On the other hand, the greedy variant selects samples without considering specific thresholds at each batch step. Although the remaining available budget may be depleted before reaching the maximum number of batch steps, very informative samples can be greedily selected early on, as long as the total available budget is not exceeded. The results[5] for this configuration are shown in Figure 4. In this case, greedy ConBatch-BAL outperforms dynamic thresholding ConBatch-BAL on the build6k and nieman17k datasets, while the opposite is true on mnist6k.

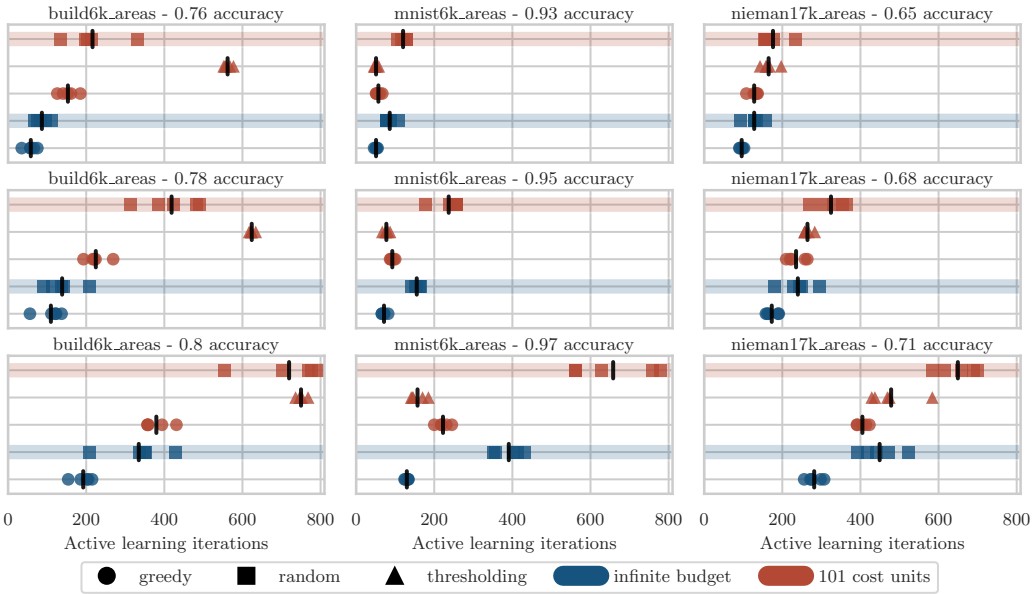

Figure 4: Benchmark results for the *area_cost* configuration across datasets and budget constraints. Markers represent the number of active learning iterations required to reach model accuracy targets on the test set for each seed, with the mean indicated by a vertical line. *Dynamic thresholding ConBatch-BAL* underperforms *greedy ConBatch-BAL*, particularly on the build6k dataset.

---

[5]The corresponding active learning curves are shown in Figure 11 in Appendix C.2

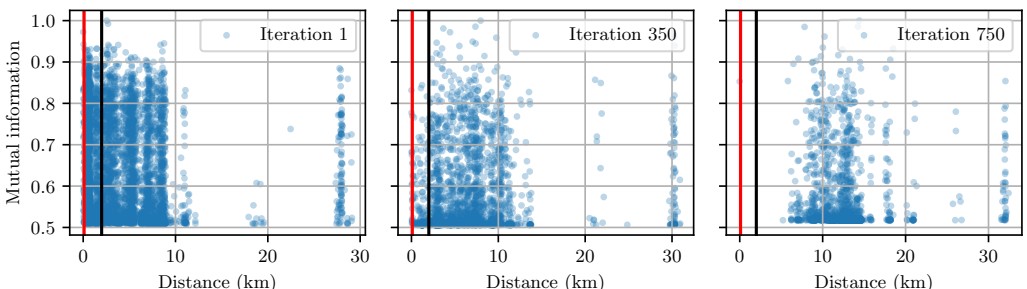

Figure 5: Normalized mutual information as a function of geographical distance for candidate samples at three distinct active learning iterations on the build6k dataset. Many informative points can be found within 2 km (black line), while fewer buildings are available within 100 m (red line).

In the *cost_area* configuration, many informative buildings are concentrated in a geographical area with high acquisition costs, making it difficult for dynamic thresholding to select informative samples under the imposed threshold constraints. In contrast, mnist6k is more diverse across areas, allowing dynamic thresholding to be competitive and even outperform greedy ConBatch-BAL.

## 7 DISCUSSION AND LIMITATIONS

The experiments indicate that the proposed ConBatch-BAL strategies can reduce the number of active learning iterations needed to reach satisfactory model accuracy in real-world scenarios with batch acquisition under budget constraints. While the average distance traveled without budget constraints is approximately 25 km per batch, as shown in Appendix C.4, similar accuracy can be achieved under the 2-km constraint across all datasets on the *distance* configuration. This finding is important for the practical adoption of active learning methods. As shown in Figure 5, informative samples are generally found within a 2 km distance of the reference sample, though fewer samples are naturally available under the stricter 100-m distance constraint. While sample informativeness depends on the data distribution and is case-specific, the effectiveness of active learning methods should not significantly reduce as long as enough informative samples can be found within the constrained region. To examine the samples collected by the tested strategies over active learning iterations, Figure 6 represents the mutual information from acquired batches of samples. More informative batches are generally collected after a few learning iterations as the training set becomes less sparse, peaking at the point where highly informative samples start becoming less available. As expected, constraints hinder the acquisition of informative samples, particularly under the 100-m batch distance limit, where the number of samples per batch is significantly compromised.

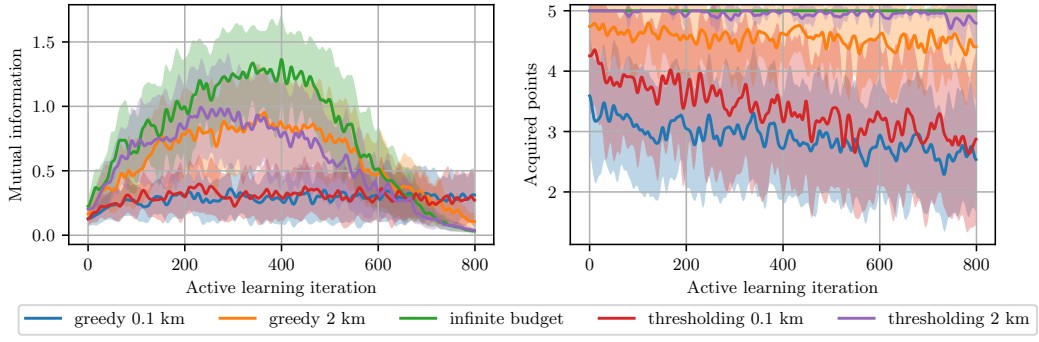

Figure 6: Mutual information for batches collected by ConBatch-BAL strategies on the *build6k* dataset. Budget constraints limit the acquisition of informative samples and result in fewer points collected per batch, especially for greedy ConBatch-BAL.

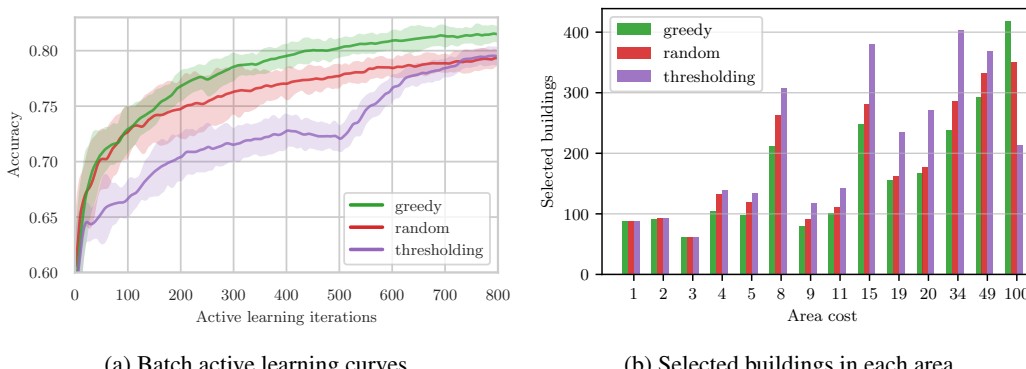

(a) Batch active learning curves.       (b) Selected buildings in each area.

Figure 7: Results for the *area_cost* configuration on the *build6k* dataset under a budget constraint of 101 cost units. (Left) *Dynamic thresholding* underperforms *greedy ConBatch-BAL* and *random selection* strategies. (Right) Greedy ConBatch-BAL selects costly but informative buildings.

The active learning curve for the *area_cost* under a constraint of 101 cost units is shown in Figure 7a. The results reveal that the main drawback of dynamic thresholding ConBatch-BAL compared to the greedy variant is its inability to acquire expensive but highly informative samples due to the imposed thresholds. In some cases, acquiring fewer but informative samples proves very effective. Figure 7b examines in which area samples are acquired throughout the experiment, showing that dynamic thresholding limits acquisition from the most expensive area compared to the other strategies. Since highly informative buildings are located in this area, the performance of dynamic thresholding is compromised. However, this effect corresponds to the extreme case where informative samples are concentrated in a region with very high acquisition costs.

### 7.1 LIMITATIONS

**Approximate uncertainty estimates.** In this work, uncertainty metrics are computed through Monte Carlo dropout BNNs. While these provide a framework for efficiently capturing the uncertainty in model predictions, the resulting uncertainty estimates might not be as informative due to the noise introduced by the MC-dropout variational approximation. Future research efforts could apply ConBatch-BAL strategies using uncertainty metrics computed through other BNN variants, such as stochastic gradient MCMC (Welling & Teh, 2011; Chen et al., 2014; Zhang et al., 2019).

**Model re-training.** While the pre-trained foundation model DINOv2 is used to compute feature embeddings from aerial imagery, we do not fine-tune DINOv2 in our experiments and only re-train the BNN classifier, thereby reducing computational demands. Although only marginal gains are reported in the literature when fine-tuning the internal layers of DINOv2 (Oquab et al., 2023), this research path could be explored further.

**Aerial imagery dataset.** In the introduced real-world building datasets, energy efficiency categories and typologies are informed by building aerial imagery. Complementing the dataset with street-view images or other informative features may lead to more accurate predictions (Mayer et al., 2023).

## 8 CONCLUSIONS

We introduce two batch active learning strategies for acquiring informative samples under budget constraints and showcase their effectiveness on real-world datasets for building energy efficiency prediction and typology classification. While solving exactly the combinatorial problem of acquiring the most informative batch under budget constraints is often intractable due to system-level statistical and cost effects, heuristic strategies like dynamic thresholding or greedy selection can offer satisfactory performance in practice. Our results show that the developed ConBatch-BAL strategies can reduce active learning iterations and cost in real-world settings with expensive data annotation and geospatial constraints, even surpassing the performance of the unconstrained random selection baseline.

REPRODUCIBILITY STATEMENT

To facilitate the reproduction of the experiments presented in Section 6 and Appendix C, we provide the *source code* as supplementary material on the submission platform. Additionally, the *real-world building datasets* introduced in Section 5, along with JSON files containing the results from the experiments, can be accessed via the following anonymous link. To comply with double-blind review requirements, the source code and the data repository have been anonymized. Instructions for reproducing the results and downloading the datasets can be found in the main README file within the source code. Configuration files containing hyperparameters and random seeds are also provided alongside the results. Minor variations in results may be expected on different machines, even if the same random seed is set. All active learning experiments presented in the paper were run on an Intel Xeon compute node with 4 CPUs and 10 GB of RAM. The computational time to complete each experiment ranged from 6 to 12 hours.

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

## A  BUILDING DATASETS CURATION PROCESS

We leverage open data sources and web services available in the Netherlands to create the released building datasets. Specifically, we retrieve aerial images via the PDOK web service (PDOK, 2024), building footprints and typologies from 3DBAG (BAG, 2024), and building energy performance data from RVO (RVO, 2024), in collaboration with the municipality of Rotterdam. To integrate building energy performance and typology data with geographical information and building foot-prints, we utilize the unique building identifier assigned to each building in the Netherlands. 8-cm resolution aerial images are collected via PDOK (PDOK, 2024) by defining the geographic region with rectangular bounding boxes that enclose the building footprints, with an added buffer to capture the surrounding context. Each aerial image is then linked to its corresponding energy efficiency or typology class by cross-referencing tabular building data and footprint through the unique building identifier. Since geographical information is available together with building footprints, we store the geolocation of each building as the centroid of its bounding box. The dataset curation process is illustrated in Figure 8.

**build6k** We randomly select approximately 6,000 buildings with an equal representation of energy-efficient and energy-inefficient classes. Energy-efficient buildings have energy labels from A to E, while energy-inefficient buildings correspond to energy labels from F to G.

**nieman17k** We randomly select approximately 17,000 buildings, balanced across the following ty-pology classes: {upstairs apartment (<1945), terraced house (<1945), terraced house (>1945), porch house (<1945), porch house (1945-1975), porch house (>1975), detached house}. These typologies correspond to Nieman categories, commonly used for building energy performance anal-ysis (Koezjakov et al., 2018).

**mnist6k** We select the same number of samples from the MNIST dataset as in the build6k dataset, ensuring they are balanced across classes. Each digit in the *mnist6k* dataset is linked to the ge-olocation of a building from the *build6k* dataset. In this modified version of MNIST, we assume annotators must travel across Rotterdam to label the images with the correct digits.

Additionally, we provide (i) building embeddings computed with the foundation model DINOv2 (Oquab et al., 2023), (ii) building geolocations, and (iii) predefined training, test, and pool split for all released datasets[6], which are listed in Table 1.

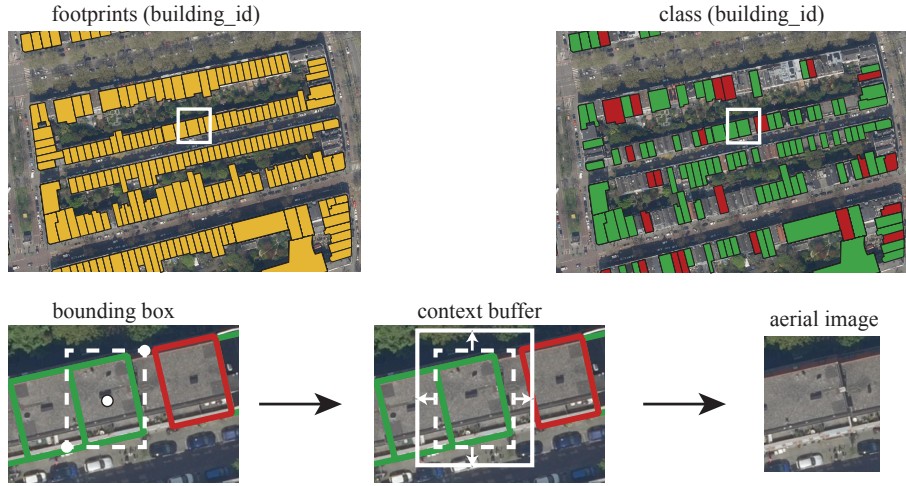

Figure 8: Datasets curation process. (Top) Building footprints and tabular data are cross-referenced using the unique identifiers assigned to the buildings. (Bottom) 8-cm resolution aerial images are collected from PDOK (PDOK, 2024) by defining a rectangular box that encloses each building footprint, including an additional buffer to capture the surrounding context.

---

[6]The datasets, *build6k* and *nieman17k*, are publicly released under a CC by 4.0 license. Anonymous link.

Table 1: Samples, classes, and predefined training, test, and pool splits for all released datasets.

| DATASET | CLASSES | SAMPLES | TRAINING SET | TEST SET | POOL SET |
|---------|---------|---------|--------------|----------|----------|
| build6k | 2 | 5999 | 30 | 1500 | 4469 |
| nieman17k | 7 | 17500 | 70 | 3500 | 13930 |
| mnist6k | 10 | 5999 | 30 | 1500 | 4469 |

Table 2: Hyperparameters set for the Bayesian neural network classifier and active learning task.

| HYPERPARAMETER | VALUE | HYPERPARAMETER | VALUE |
|----------------|-------|----------------|-------|
| Hidden layers | 2 | Layer width | 256 |
| Optimizer | Adam | Learning rate | 0.0001 |
| Batch size | 32 | Weight decay | 0.0001 |
| Epochs | 200 | Dropout rate | 0.1 |
| Forward passes | 100 | Active batch steps | 5 |

## B  IMAGE EMBEDDINGS AND HYPERPARAMETERS

### B.1  EMBEDDING VECTORS COMPUTED WITH DINOv2

Before running our active learning experiments, we precompute the 384-dimensional embedding vector[7] from each aerial or digit image using the vision transformer DINOv2 ViT S/14 with registers (Oquab et al., 2023). As shown in Figure 9, the Bayesian neural network classifiers retrained in our active learning experiments take image embeddings as input and output class probabilities, randomly generated through Monte Carlo dropout, as detailed in Section 3.3.

### B.2  HYPERPARAMETERS

The hyperparameters set for the Monte Carlo dropout Bayesian neural network (BNN) are listed in Table 2, fine-tuned on the dataset *build6k* under an infinite budget. The hyperparameters remain unmodified as the classifier is retrained at each active learning iteration. As retraining the classifier does not involve long computations, we can afford retraining the BNN model from scratch. Note that the input embedding vector size is 384, corresponding to the dimensionality of the features provided by DINOv2, as described in Appendix B.1. In each experiment, we run 800 active learning iterations, collecting batches of 5 points at each iteration. After each batch is acquired, the BNN model is retrained.

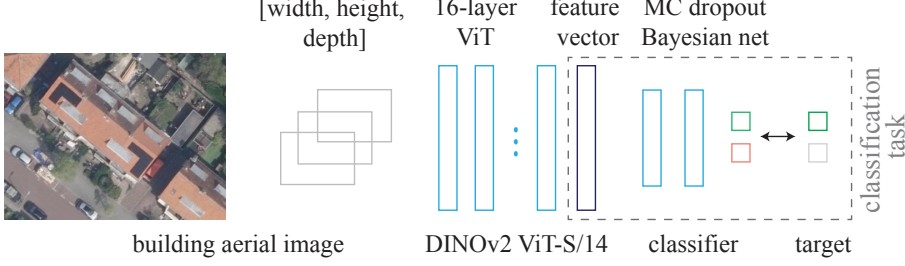

Figure 9: Pipeline: from aerial images and embeddings obtained by the foundation model DINOv2 to class probabilities generated by a Monte Carlo dropout Bayesian neural network.

---

[7]The generated image embeddings for all datasets are also publicly released. Anonymous link.

# C ADDITIONAL RESULTS

## C.1 DISTANCE_RETURN CONFIGURATION

The results across all introduced datasets for the *distance_return* cost configuration are shown in Figure 10, representing the number of active learning iterations required to achieve a specified model accuracy target on the test set. The trends are similar to those reported in Figure 3 for the *distance* cost configuration, with ConBatch-BAL strategies outperforming the random selection baseline in all tested settings. The main difference is that more iterations are required in this configuration, as the constraints are stricter due to the requirement of returning to the initial selected point. Interestingly, the proposed ConBatch-BAL strategies under the 100-m batch constraint require fewer iterations than the unconstrained random selection baseline to achieve a model accuracy of 97% on the *mnist6k* dataset.

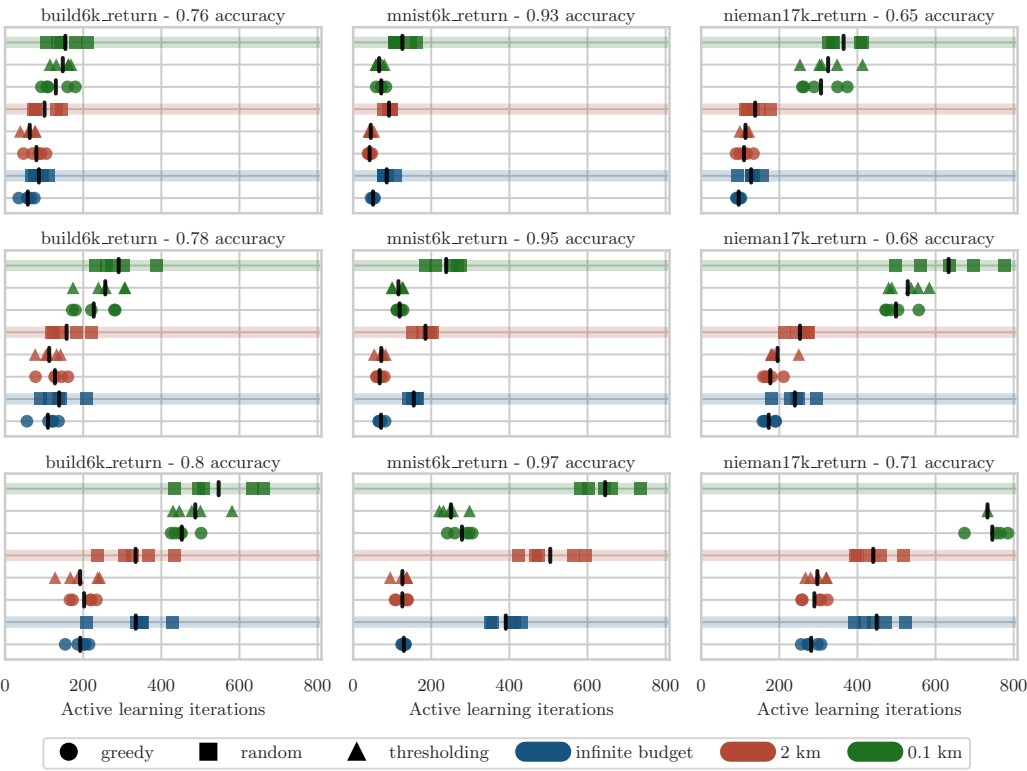

Figure 10: Benchmark results for the *distance_return* cost configuration across datasets and budget constraints. Markers represent the number of active learning iterations required to reach a target model accuracy on the test set for each seed, with the mean indicated by a vertical line. *Greedy ConBatch-BAL* and *dynamic thresholding ConBatch-BAL* outperform the random selection baseline. More active learning iterations are required to reach the model accuracy targets compared to the results for the *distance* configuration, shown in Figure 3. Under the 100-meter constraint, the random selection baseline fails to achieve the 71% model accuracy target on the nieman17k dataset within 800 active learning iterations.

## C.2  ACTIVE LEARNING CURVES

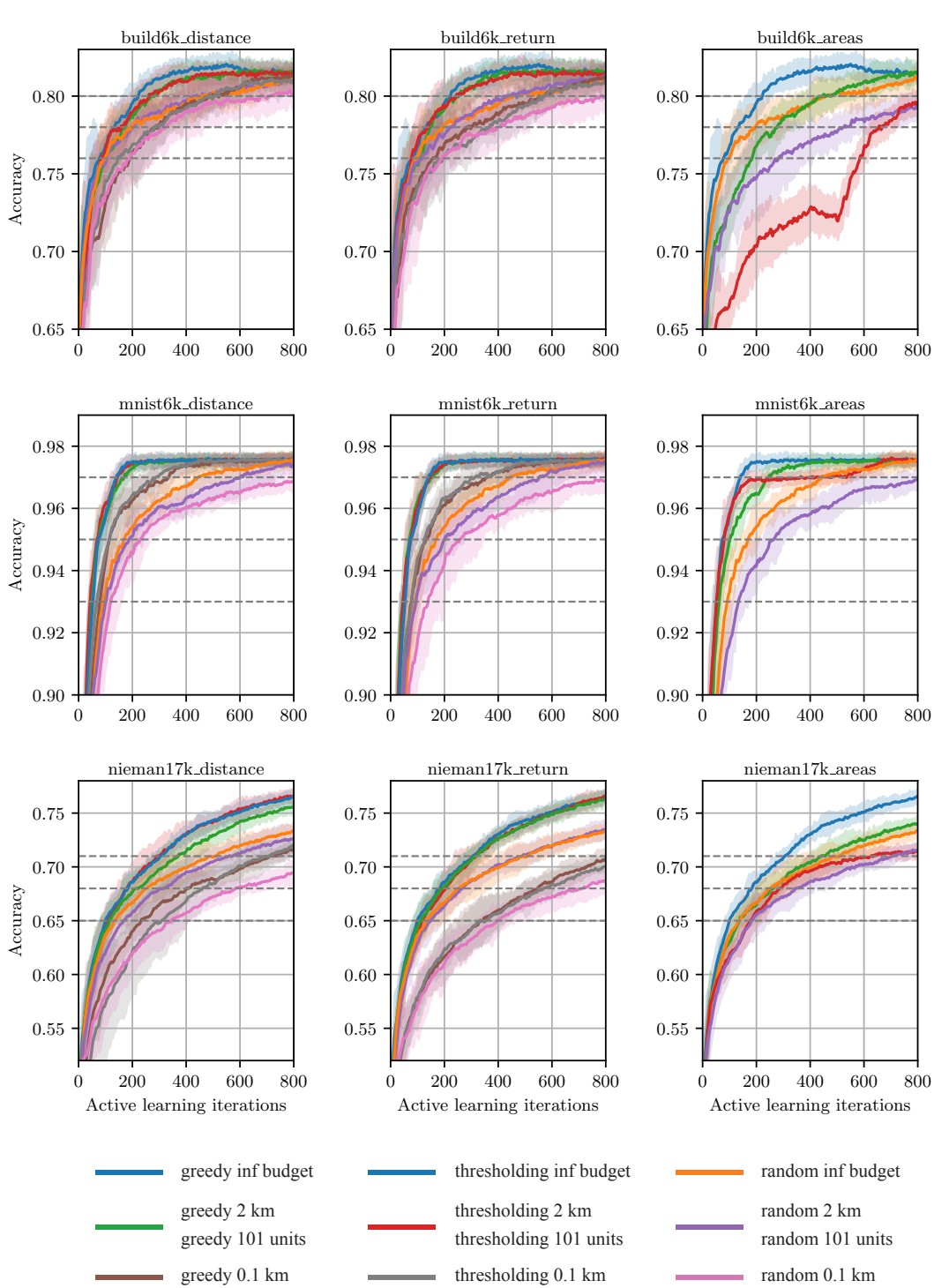

Figure 11: Active learning curves for all tested strategies across datasets and budget constraints. Model accuracy on the test set is shown over active learning iterations, with the mean over 5 seeds represented by a solid line, and the shaded area indicating two standard deviations. Dashed lines mark the accuracy targets defined in Figures 3, 4, and 10

## C.3 ACQUIRED SAMPLES AND ACQUISITION COST

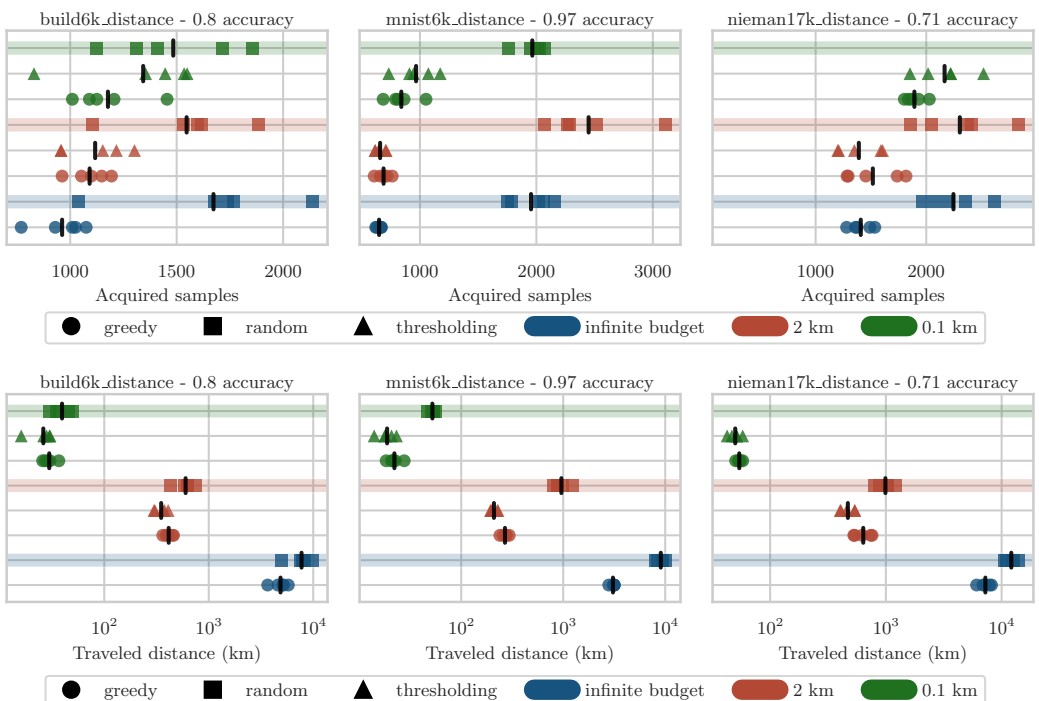

Figure 12: Benchmark results for the *distance* configuration, showcasing the number of acquired samples and acquisition cost across datasets and budget constraints. Markers represent (top) the total number of acquired samples and (bottom) the traveled distance required to reach a target model accuracy on the test set for each seed, with the mean indicated by a vertical line.

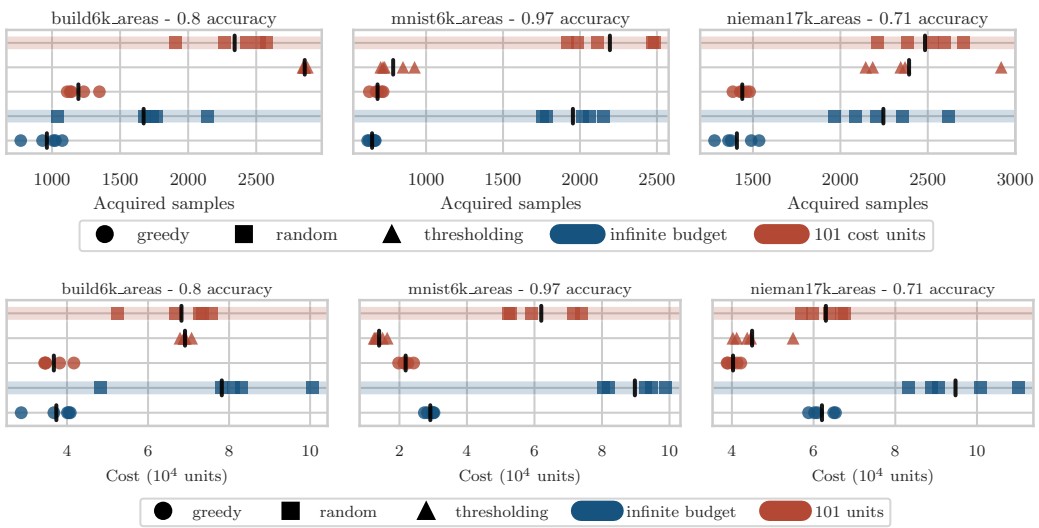

Figure 13: Benchmark results for the *area_cost* configuration, showcasing the number of acquired samples and acquisition cost across datasets and budget constraints. Markers represent (top) the number of acquired samples and (bottom) the cost required to reach a target model accuracy on the test set for each seed, with the mean indicated by a vertical line.

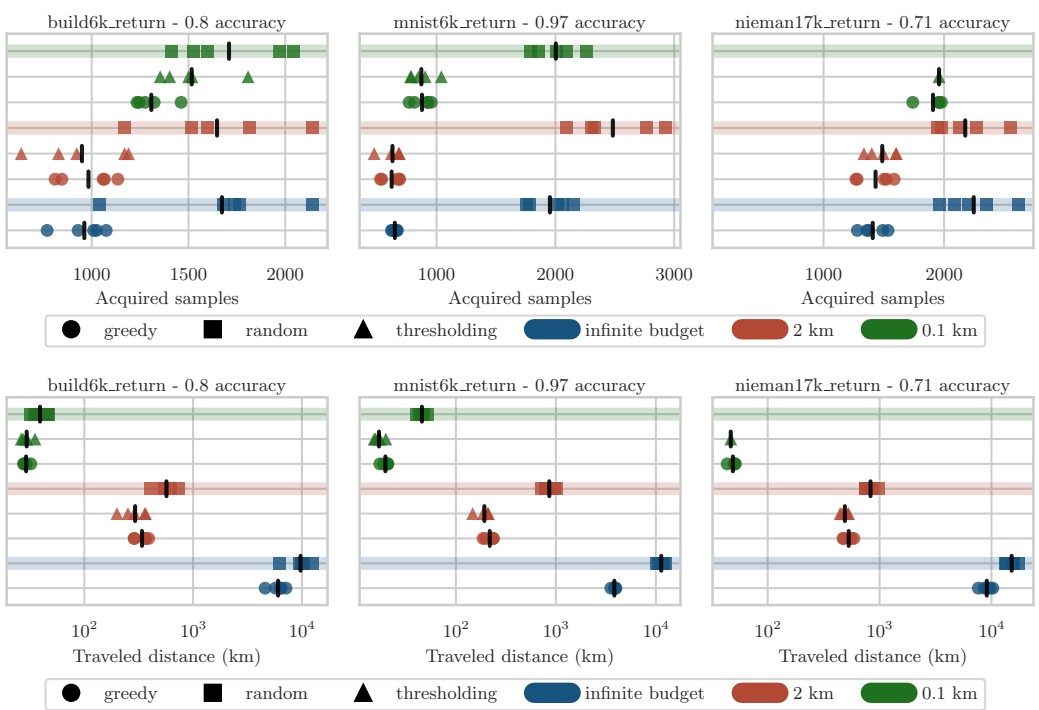

Figure 14: Benchmark results for the *distance_return* configuration, showcasing the number of acquired samples and acquisition cost across datasets and budget constraints. Markers represent (top) the total number of acquired samples and (bottom) the traveled distance required to reach a target model accuracy on the test set for each seed, with the mean indicated by a vertical line.

## C.4 ACQUISITION COST AND NUMBER OF ACQUIRED POINTS PER BATCH

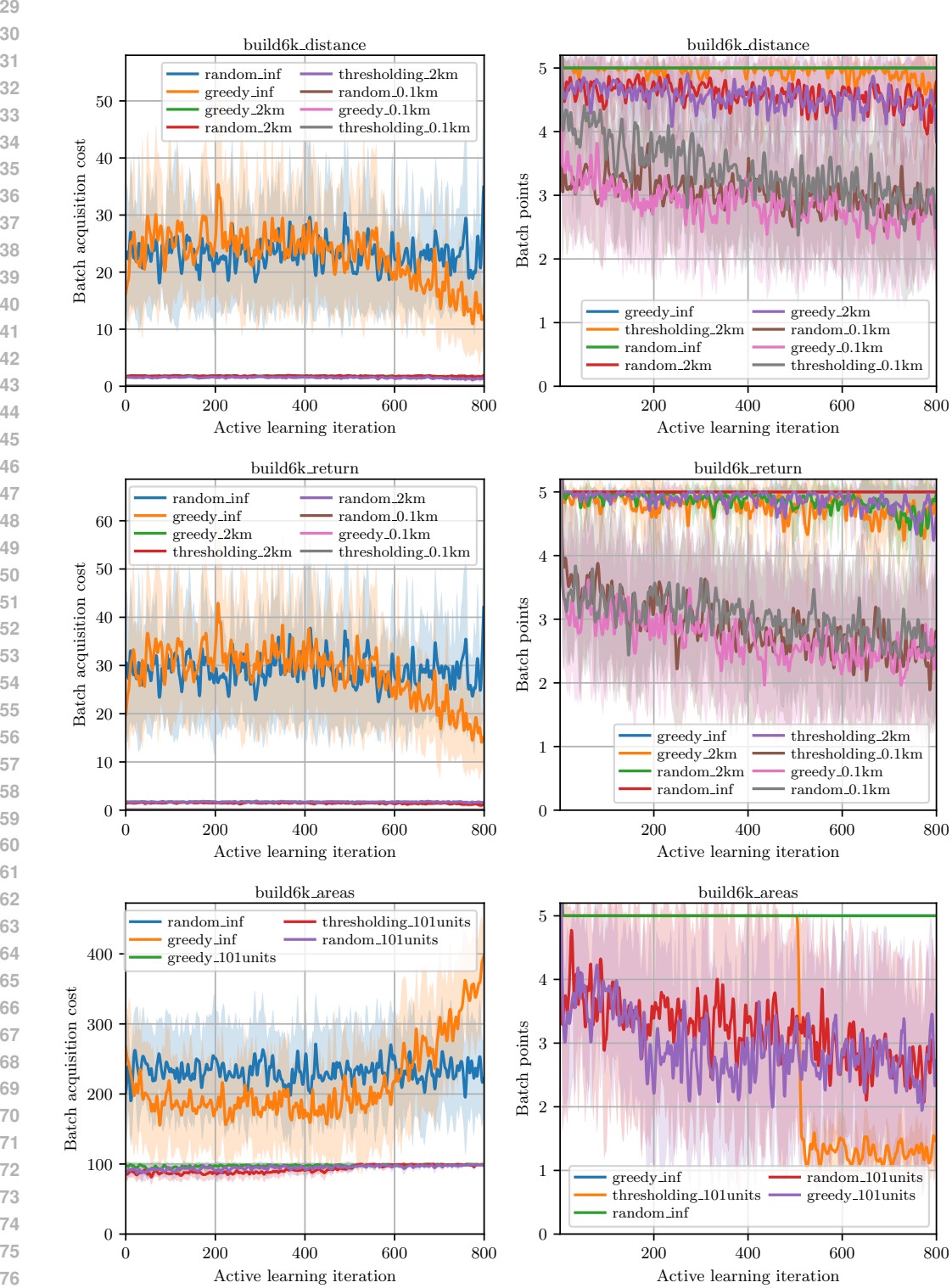

Figure 15: Batch acquisition cost (left) and acquired points (right) at each active learning iteration for all tested cost configuration and strategies on the *build6k* dataset. In all subfigures, the mean over 5 seeds is represented by a solid line with the shaded area indicating one standard deviation.

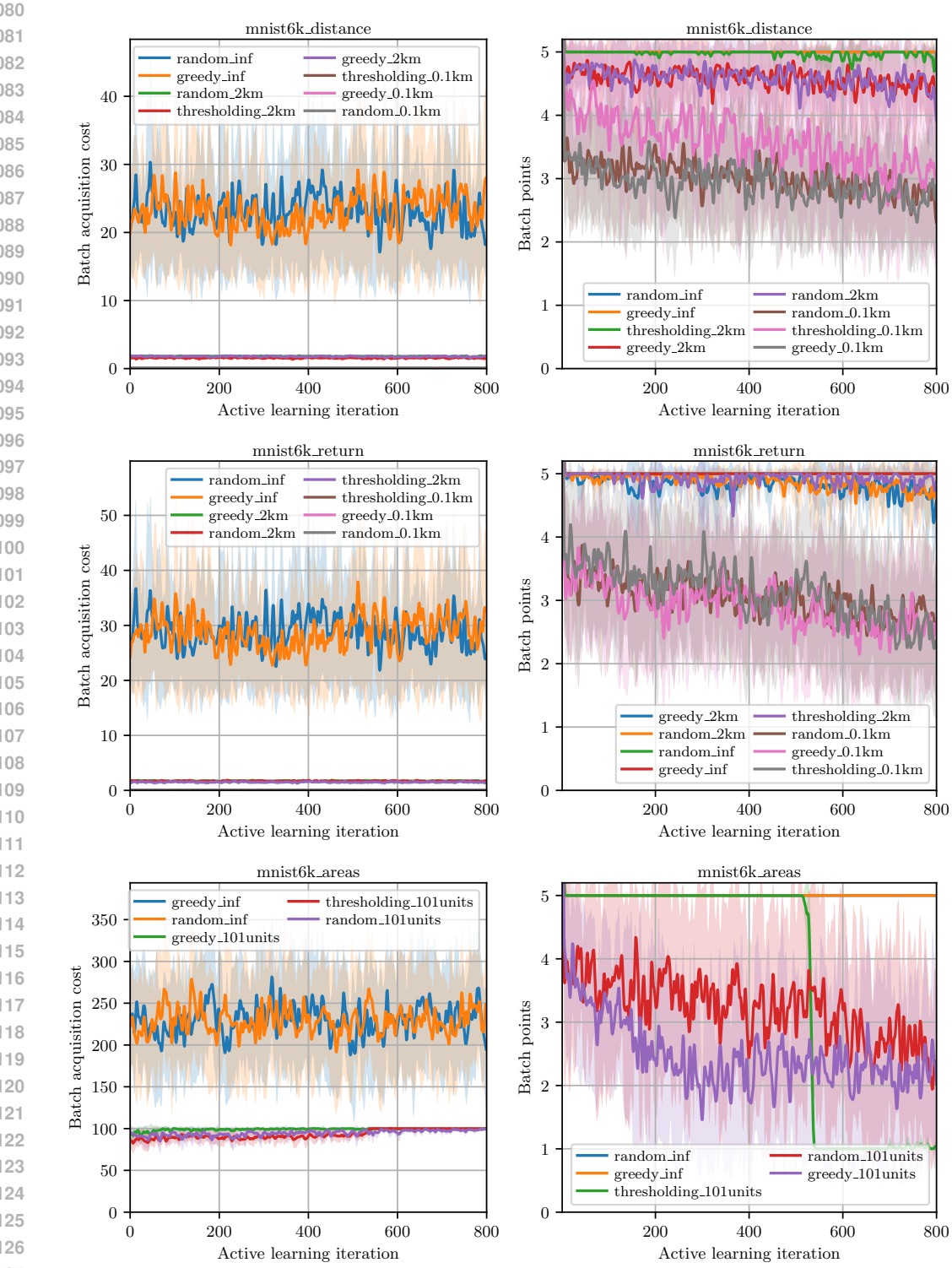

Figure 16: Batch acquisition cost (left) and acquired points (right) at each active learning iteration for all tested cost configurations and strategies on the *mnist6k* dataset. In all subfigures, the mean over 5 seeds is represented by a solid line with the shaded area indicating one standard deviation.

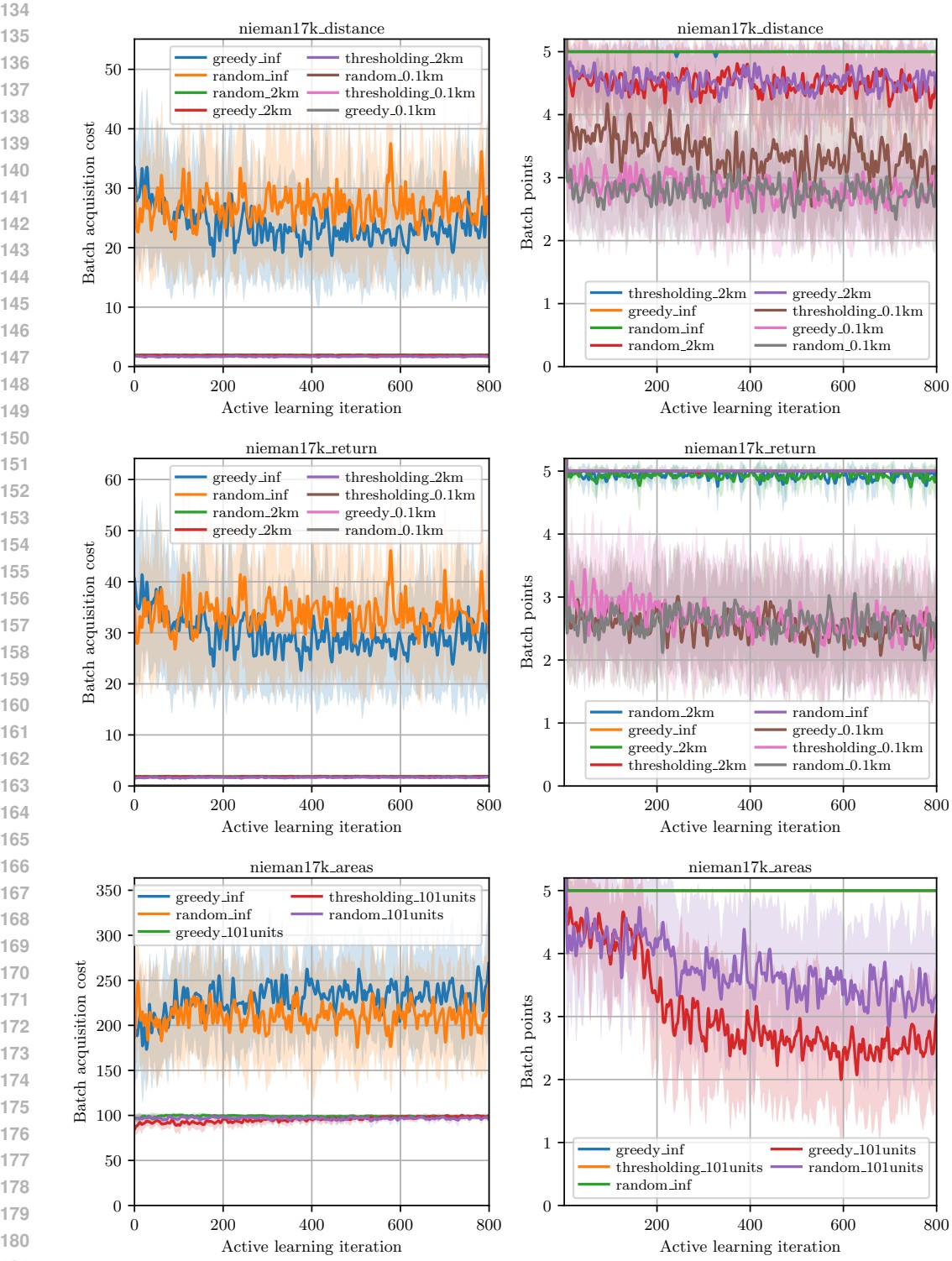

Figure 17: Batch acquisition cost (left) and acquired points (right) at each active learning iteration for all tested cost configurations and strategies on the *nieman17k* dataset. In all subfigures, the mean over 5 seeds is represented by a solid line with the shaded area indicating one standard deviation.

## C.5 BATCH SIZE

To evaluate the impact of the batch size, $n_{\max}$, on the performance of the proposed strategies, we conduct additional experiments on the *build_6k* dataset and the *distance* configuration with batch sizes of 2 and 10. The results are presented in Figure 18, representing the required number of active learning iterations or acquired samples to reach a model performance of 80% on the test set. Additionally, the corresponding learning curves are shown in Figure 19.

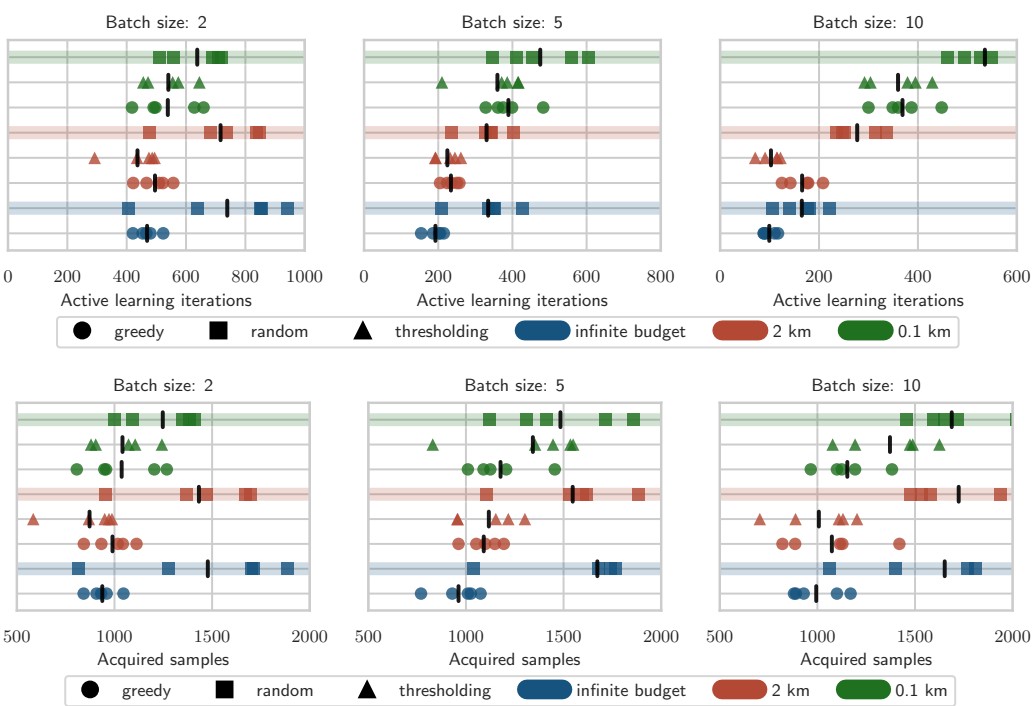

Figure 18: Benchmark results for the *build_6k* dataset under the *distance* configuration, with varying budget constraints and batch sizes (2, 5, and 10). Markers represent (top) the number of active learning iterations and (bottom) the total number of acquired samples required to reach a target model accuracy of 80% on the test set for each seed, with the mean indicated by a vertical line.

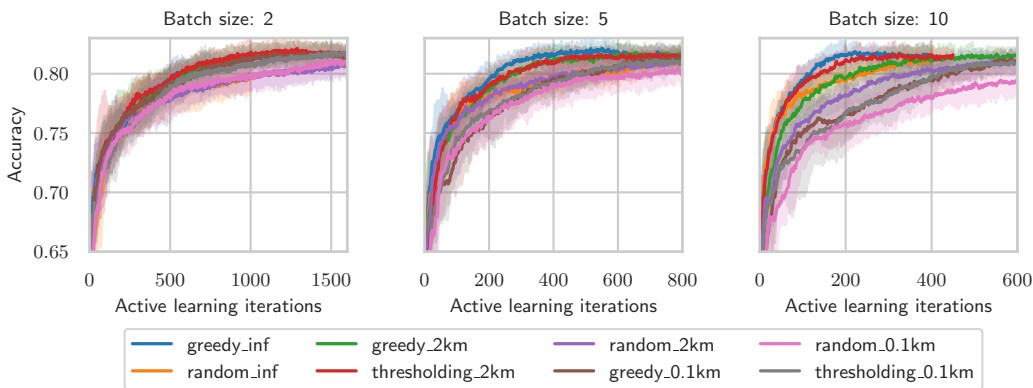

Figure 19: Active learning curves for the *build_6k* dataset under the *distance* configuration, with varying budget constraints and batch sizes (2, 5, and 10). Model accuracy on the test set is shown over active learning iterations, with the mean over 5 seeds represented by a solid line, and the shaded area indicating two standard deviations.

The effectiveness of ConBatch-BAL strategies relative to the baseline remains consistent across batch sizes, outperforming random selection in all tested settings. While more active learning iterations are required to reach the accuracy target for smaller batch sizes, this is naturally caused by the fewer number of samples that can be acquired per tour. To compare the strategies across different batch sizes, we examine the number of acquired samples required to achieve the accuracy target. The results indicate that, in most cases, the number of acquired samples increases with higher batch sizes. We attribute this result to the fact that the overall available budget per sample decreases for higher batch sizes. However, dynamic thresholding ConBatch-BAL experiences less variation over batch sizes, as it balances the budget across the batch.

