# OpenReview forum: "ConBatch-BAL: Batch Bayesian Active Learning under Budget Constraints"
_ICLR.cc/2025/Conference — Submitted to ICLR 2025_

### Official Review · Reviewer_pqZH · 2024-10-25

**Soundness:** 2
**Presentation:** 3
**Contribution:** 2
**Rating:** 3
**Confidence:** 4

**Summary:**

This paper introduces two Bayesian active learning heuristics—greedy acquisition and dynamic thresholding—for batch acquisition under budget or cost constraints. The authors conducted experiments on one artificial dataset derived from MNIST and two real-world datasets, demonstrating improved performance.

**Strengths:**

The authors clearly presented the two heuristics.

**Weaknesses:**

(1) Vague problem setting: The authors do not provide a clear objective or constraints in Section 3.1. For instance, when considering the examples in Figure 2(b), the cost of a batch depends on the order of the batches, which implies that in Equation 2, the constraints should be $c(x_1, ..., x_n) \leq c_{max}, n \leq n_{max}$. The combinatorial nature of the cost is overlooked in the problem setting.

(2) Limited applications: The proposed methods are applied to only two real-world datasets, both of which are aerial imagery, limiting the scope of the applications.

(3) Limited Bayesian Neural Network (BNN) models tested: The experiments focus solely on MC-Dropout, but other BNN models with superior performance—such as SGHMC [1], SG-MCMC [2], and cSG-MCMC [3]—should also be evaluated.


references:

[1] Chen, Tianqi, Emily Fox, and Carlos Guestrin. "Stochastic gradient hamiltonian monte carlo." In International conference on machine learning, pp. 1683-1691. PMLR, 2014.

[2] Welling, Max, and Yee W. Teh. "Bayesian learning via stochastic gradient Langevin dynamics." In Proceedings of the 28th international conference on machine learning (ICML-11), pp. 681-688. 2011.

[3] Zhang, Ruqi, Chunyuan Li, Jianyi Zhang, Changyou Chen, and Andrew Gordon Wilson. "Cyclical stochastic gradient MCMC for Bayesian deep learning." arXiv preprint arXiv:1902.03932 (2019).

Minor suggestions:

(1) In line 134, use ${x_1, ..., x_n} \subset D_{pool}$.

(2) Ensure consistency in the use of "Batch-BALD" or "batch-BALD" (see line 149 vs. line 346).

(3) There is no need to use both "return" and "output" with the same content simultaneously in Algorithm 1 and Algorithm 2.

**Questions:**

It appears that the two strategies, which are the main contributions of this paper, can be easily extended to classical active learning algorithms, such as CoreSet or BADGE. Even in a Bayesian setting, average embeddings can still be used for active learning with CoreSet or BADGE. I believe the main technical challenge lies in addressing the combinatorial cost. The authors should avoid restricting their approach to Bayesian Neural Networks (BNNs) without offering additional benefits beyond BNN model accuracy. I am open to discussing this further in the future.

---

> ### Author Response · Authors · 2024-11-22
> **Response to Reviewer pqZH**
>
> Thank you for taking the time to review our paper. We have carefully addressed each of your comments below.
>
> >**W1:** Vague problem setting: The authors do not provide a clear objective or constraints in Section 3.1...
>
> In the paper, we highlight the complexities involved in the combinatorial sample selection problem under budget constraints (Lines 184 – 188). These challenges motivated us to introduce two new heuristic strategies. We understand, however, that the reviewer’s concern may be also caused by the notation.
>
> ```Based on this comment, we have revised the notation in the updated manuscript (Section 3.1, Eq. 2). We have also explained that the acquisition cost is defined as a function of an ordered sequence of samples (Lines 137-139).```
>
> >**W2:** Limited applications: The proposed methods are applied to only two real-world datasets, both of which are aerial imagery, limiting the scope of the applications.
>
> The proposed strategies cannot be directly tested on traditional active learning datasets, unless a synthetic cost model and relevant cost constraints are defined. In the paper, we test the proposed active learning strategies on two new real-world datasets, which we believe is an important contribution to the field. Based on the conducted experiments, we can draw practical insights. For example, we point out that a high prediction accuracy can be achieved under strict budget constraints in the tested real-world scenarios.
>
> Additionally, our paper also includes a synthetic dataset derived from MNIST, in which each digit is associated with a geolocation, in order to test the explored active learning strategies on a classical dataset. While our contributions constitute a step forward toward the application of active learning in real-world applications, we encourage the further development of novel real-world datasets inspired by our work and problem formulation. A more detailed explanation can be found in the global response **G1**.
>
> >**W3:** Limited Bayesian Neural Network (BNN) models tested...
>
> The main scope of our paper is the problem formulation of active learning strategies for real-world problems under budget constraints, including the development of two strategies that rely on an acquisition function derived from mutual information. While testing alternative Bayesian neural network (BNN) variants is certainly a valuable research direction, we selected Monte Carlo dropout because it is more computationally efficient than other inference approaches, as explained in Lines 041-047. Exploration in other BNN directions is limited by the scale of our problem and time limitations of this review process. But it might also deserve a standalone publication to be done properly. We also encourage the investigation of alternative BNN approaches in Section 7.1.
>
> ```Although we were not able to implement, fine-tune, and evaluate alternative BNN approaches, we have included the reviewer’s suggested works in the revised manuscript as potential BNN methods for future research (Lines 518-520).```
>
> >**Suggestions:** (1) In line 134, use x1,...,xn⊂Dpool.  (2) Ensure consistency in the use of "Batch-BALD" or "batch-BALD" (see line 149 vs. line 346).  (3) There is no need to use both "return" and "output" with the same content simultaneously in Algorithm 1 and Algorithm 2.
>
> ```We thank the reviewer for the suggested edits. Based on this comment, we have corrected the notation in the revised manuscript (Lines 134, 222, 231, 240, 248, 333, 346).```
>
> >**Q1:** It appears that the two strategies, which are the main contributions of this paper, can be easily extended to classical active learning algorithms, such as CoreSet or BADGE...
>
> This suggestion is truly interesting. The active learning strategies proposed in this paper could be integrated with other frameworks, such as CoreSet or BADGE, which utilize gradient embeddings and clustering to sequentially select samples for annotation. As noted in **W2**, this work represents a step forward in applying active learning to real-world problems under budget constraints, while also opening up directions for future research, including the investigation of alternative acquisition functions.

---

> > ### Comment · Reviewer_pqZH · 2024-11-26
> > **Further comments**
> >
> > I appreciate the authors' response. However, I feel that my concerns were not addressed or adequately considered. Therefore, I will maintain my original score.

---

### Official Review · Reviewer_NNgV · 2024-10-31

**Soundness:** 2
**Presentation:** 2
**Contribution:** 2
**Rating:** 3
**Confidence:** 4

**Summary:**

This paper tackles the challenge of batch Bayesian active learning under budget constraints by introducing two novel heuristics: dynamic thresholding and greedy strategies. It also presents two new real-world datasets featuring geolocated aerial images of buildings. Experimental results show that these strategies outperform a random selection baseline across both real-world datasets and a geolocated MNIST dataset.

**Strengths:**

The paper addresses the practical challenge of batch Bayesian active learning under constraints on both batch size and cost. The two real-world datasets introduced offer valuable resources for evaluating active learning solutions with geolocated points.

**Weaknesses:**

The paper would benefit from a clearer and more detailed description of the method and an improvement of the technical contributions.

1. While the objective is to achieve high accuracy on the test set, the test set itself has not been utilized in the the solution. If the goal is to improve test set accuracy, why not consider incorporating mutual information with the classification of points in the test set, similar to the approach in the well-known work by Andreas Krause, Near-Optimal Sensor Placements in Gaussian Processes (2008)? In that study, the test set can be treated as the entire design space.

2. The main technical contribution, the two active learning strategies, is explained too briefly and lacks essential details. For instance, the description omits the acquisition function selecting each point in the batch sequentially (before showing it in the pseudocode), as well as the expression of mutual information using Dropout samples. Lines 185–188 raise several challenges, but there is no clear explanation of how these issues are addressed. Additionally, there are notation inconsistencies in Algorithms 1 and 2:

    + Algorithm 1: $\\mathbf{x}\_1^*, \\dots, \\mathbf{x}\_n^*$ and $n$ are used (lines 222, 231) before they are defined. Line 222 could be revised to "return $A\_{i-1}$" and Line 231 to "return $A\_{n\_{\\max}}$". Similar issues exist in Algorithm 2.

    + Line 225: argmax can be revised to argmax_x.

    + $c(x)$ depends on the previous selected point so the input to $c$ should include the previous selected point.

    + What is the computational complexity of the approach?

3. The heuristic employed is relatively straightforward: the algorithm divides the budget constraint by the batch size or aims to fully utilize the available budget with each sampled point.

4. The experiments have several weaknesses:

+ Why not plot accuracy over iterations, instead of only showing the number of iterations required to reach selected accuracy levels?
+ The experiment includes only a simple baseline (random selection), while other batch Bayesian active learning approaches could also be evaluated under an unlimited budget. For a finite budget, some selected points could simply be dropped from the batch to stay within budget.
+ There is no clear description of the training set, test set, initial training set (before active learning), or the number of Dropout samples.
+ The impact of batch size on algorithm performance is not explored.

Additional notation issues:

+ $y\_1,\\dots,y\_n$ is used to denote the outputs of $\\mathbf{x}\_1, \\dots, \\mathbf{x}\_n$, but in lines $172$ $y\_{c\_1}, \\dots, y\_{c\_k}$ refer to the $k$ classes.

+ Line 318 states that the "cost is inversely proportional to the number of buildings," yet it also notes that "crowded areas are more expensive," which seems contradictory.

**Questions:**

Please review the weaknesses mentioned above.

---

> ### Author Response · Authors · 2024-11-22
> **Response to Reviewer NNgV (1/2)**
>
> Thank you for taking the time to review our paper. We have carefully addressed each of your comments below.
>
> >**W1:** The paper would benefit from a clearer and more detailed description of the method and an improvement of the technical contributions.
>
> ```We have thoroughly revised the notation and descriptions of the proposed active learning strategies in the revised manuscript.```
>
> > While the objective is to achieve high accuracy on the test set, the test set itself has not been utilized in the the solution...
>
> We evaluate the model on a separate test set to ensure it performs well on unseen data. While selecting points based on the test set might improve performance on that specific set, it risks compromising generalization to new, unseen data. We acknowledge that the description of the task’s objective could be clarified in the main text.
>
> ```We now mention in the revised manuscript that the main goal is to quantify the performance of the trained model on a separate test set, which is considered unavailable during the training stage. (Lines 131-132)```
>
> >**W2:** The main technical contribution, the two active learning strategies, is explained too briefly and lacks essential details. For instance, the description omits the acquisition function selecting each point in the batch sequentially...
>
> In our paper, the formulation for computing mutual information is presented in Section 3.2 (Eqs. 3 and 4). Due to space limitations, we direct the reader to the relevant references for additional details.
>
> >Additionally, there are notation inconsistencies in Algorithms 1 and 2:
> **W2.1:** Algorithm 1: x1∗,…,xn∗ and n are used (lines 222, 231) before they are defined. Line 222 could be revised to "return Ai−1" and Line 231 to "return Anmax". Similar issues exist in Algorithm 2.
>
> ```In response to this comment, we have corrected the manuscript accordingly (Lines 222, 231, 240, 248).```
>
> >**W2.2:** Line 225: argmax can be revised to argmax_x.
>
> ```In response to this suggestion, we have corrected the manuscript accordingly (Line 225, 243).```
>
> >**W2.3:** c(x) depends on the previous selected point so the input to c should include the previous selected point.
>
> Quantities $c_{th}$ and $c_{\text{max}}$ are adjusted at each loop step based on the selected points, so we can specify only $c(\mathbf{x})$ in the conditional statements: $c(\mathbf{x}) < c_{th}$ or $c(\mathbf{x}) > c_{\text{max}}$.
>
> **W2.4:** What is the computational complexity of the approach?
>
> ```In the revised manuscript, we detail the computational complexity of the proposed strategies (Lines 213-215). Note that the computational complexity is primarily dominated by the estimation of mutual information.```
>
> >**W3:** The heuristic employed is relatively straightforward: the algorithm divides the budget constraint by the batch size or aims to fully utilize the available budget with each sampled point.
>
> In our work, we present a novel problem statement for active learning under budget constraints. Given the challenges of the combinatorial selection process (outlined in Lines 180–190), heuristics provide a practical and effective solution, especially in the context of large-scale real-world applications. However, we cannot and do not claim that the proposed strategies precisely solve the combinatorial problem. Alternative approaches, such as neural combinatorial optimization, could be employed for selection, but would incur significant computational costs.
>
> >**W4:** The experiments have several weaknesses:
> **W4.1:** Why not plot accuracy over iterations, instead of only showing the number of iterations required to reach selected accuracy levels?
>
> In the main text, we represent the number of iterations needed to reach a specified model accuracy target across all seeds, highlighting the active learning benefits of the proposed strategies, such as reduced iterations (tours in our experiments). However, the accuracy over iteration curves for all experiments is reported in Appendix C2, Figure 11.
>
> >**W4.2:** The experiment includes only a simple baseline (random selection), while other batch Bayesian active learning approaches could also be evaluated under an unlimited budget...
>
> Our choice of a mutual information-based acquisition function is motivated by literature findings (e.g., BALD, Batch-BALD), which consistently report superior performance over other conventional functions, including entropy and variation ratios. Additionally, we include a random selection baseline, as it represents a common approach used in practice for auditing building energy performance. As noted in the global comment **G1**, our work introduces a novel problem statement for active learning under budget constraints, opening up future research opportunities, including exploring alternative acquisition functions.

---

> > ### Author Response · Authors · 2024-11-22
> > **Response to Reviewer NNgV (2/2)**
> >
> > >**W4.3:** There is no clear description of the training set, test set, initial training set (before active learning), or the number of Dropout samples.
> >
> > Due to space limitations, the description of the initial training, test, and pool sets are provided in Table 1 (Appendix A), and the set hyperparameters are listed in Table 2 (Appendix B). For clarification, we include the tables in this response:
> >
> > Table 1:
> >
> > | **DATASET**   | **CLASSES** | **SAMPLES** | **TRAINING SET** | **TEST SET** | **POOL SET** |
> > |---------------|-------------|-------------|------------------|--------------|--------------|
> > | build6k       | 2           | 5999        | 30               | 1500         | 4469         |
> > | niemann17k    | 7           | 17500       | 70               | 3500         | 13930        |
> > | mnist6k       | 10          | 5999        | 30               | 1500         | 4469         |
> >
> > Table 2:
> > | **HYPERPARAMETER**   | **VALUE** | **HYPERPARAMETER**    | **VALUE**  |
> > |-----------------------|-----------|-----------------------|------------|
> > | Hidden layers         | 2         | Layer width           | 256        |
> > | Optimizer             | Adam      | Learning rate         | 0.0001     |
> > | Batch size            | 32        | Weight decay          | 0.0001     |
> > | Epochs                | 200       | Dropout rate          | 0.1        |
> > | Forward passes        | 100       | Active batch steps    | 5          |
> >
> >
> > If the reviewer considers this information should be included in the main text, we can move the tables from the appendix to the main text.
> >
> > >**W4.4:** The impact of batch size on algorithm performance is not explored.
> >
> > We agree with the reviewer and recognize that analyzing the impact of maximum batch size can strengthen the experimental analysis.
> >
> > ```In response to this comment, we conduct and report a new set of experiments with varying batch sizes. The results are shown in the new Appendix C.5. The experiments show that the performance of the proposed ConBatch-BAL strategies relative to the baseline remains consistent across batch sizes.```
> >
> > >**Suggestions:** y1,…,yn is used to denote the outputs of x1,…,xn, but in lines 172 yc1,…,yck refer to the k classes. Line 318 states that the "cost is inversely proportional to the number of buildings," yet it also notes that "crowded areas are more expensive," which seems contradictory.
> >
> > ```In response to this comment, we have corrected the manuscript accordingly (Lines 171-172, 318). ```

---

> > > ### Comment · Reviewer_NNgV · 2024-11-26
> > >
> > > I would like to thank the authors for addressing my concerns and for including new experiments with varying batch sizes in the revised manuscript.
> > >
> > > Regarding the formulation of mutual information, since the approach sequentially selects each point in the batch, if the batch size is 3, there should be a mutual information expression specifically for selecting the second point given the first (a form of mutual information conditioned on previously selected points). Additionally, the mutual information formulation should explicitly incorporate the use of dropout samples, aligning with the paper’s description that dropout samples are used to approximate the predictive distribution.
> > >
> > > Unfortunately, I find the lack of sufficient theoretical justification and the simplicity of the heuristic approach unconvincing, which leads me to maintain my current score.

---

### Official Review · Reviewer_7Kd8 · 2024-11-03

**Soundness:** 3
**Presentation:** 3
**Contribution:** 3
**Rating:** 5
**Confidence:** 3

**Summary:**

This work introduces two Bayesian active learning strategies, called ConBatch-BAL, designed for batch acquisition under budget constraints. One strategy uses dynamic thresholding to redistribute the budget across the batch, while the other follows a greedy approach to select top-ranked samples within the budget. Both strategies leverage uncertainty metrics computed by Bayesian neural networks.

The study focuses on scenarios with high annotation costs and geospatial constraints, releasing two new datasets of geolocated aerial building images annotated with energy efficiency or typology classes. ConBatch-BAL strategies are benchmarked against a random acquisition baseline and demonstrate improvements, reducing active learning iterations and data acquisition costs, while outperforming the baseline in real-world settings.

**Strengths:**

Diverse datasets and interesting applications. This paper's innovations lie in introducing dynamic thresholding for addressing varying annotation costs problems in AL. This problem definition is clear and worth studying. The experiments datasets are diverse, as they are not limited to typical vision datasets (CIFAR10, CIFAR100, tinyimageNet) but use aerial images.

**Weaknesses:**

1. The authors shall provide more hyperparameter tuning/ablation study on Section 4.1 on how to choose dynamic thresholding.

The initial threshold, c_th,1 = c_max / n_max, is determined by the total budget divided by the maximum allowed batch steps. This choice is intuitive, but it may not adapt well to scenarios with highly varying annotation costs or where uncertainty scores vary significantly from step to step. Can you conduct  an ablation study on the impact of different n_max values could help understand how sensitive the method is to this parameter and provide insights into optimal settings for datasets listed in the papers?

2. The threshold is updated by subtracting the cost of the previously selected sample and dividing by the remaining steps. This is straightforward but may lead to inefficient sampling if a high-cost sample is selected early on, reducing the budget for remaining samples.

Can the authors provide theoretical insights or guarantees about the convergence or stability of this threshold adjustment process could work? For instance, under certain assumptions about cost distribution or sample uncertainty, it might be possible to guarantee that the threshold remains within certain bounds or converges to a stable value?

3. In real-world scenarios, annotation costs and uncertainty values can be highly variable. It’s unclear if the dynamic thresholding method can robustly handle such variability or if it might be affected by extreme values (outliers). Can the authors provide experiments on highly extreme values (outliers) results?

**Questions:**

No. See weakness parts.

---

> ### Author Response · Authors · 2024-11-22
> **Response to Reviewer 7Kd8**
>
> Thank you for taking the time to review our paper. We have carefully addressed each of your comments below.
>
> >**W1:** The authors shall provide more hyperparameter tuning/ablation study on Section 4.1 on how to choose dynamic thresholding...
>
> We agree with the reviewer that the dynamic thresholding-based strategy may be less effective in certain settings, such as scenarios where informative points are very costly. This limitation is acknowledged in the paper. We have identified a specific case study (cost_area) where adaptation of dynamic thresholding is lacking, and we quantify and explain it (Lines 395-405). Given the complexity of the combinatorial selection problem, computing the initial budget threshold based on the total budget and the number of batch points is an intuitive approach that has proven effective on the tested real-world datasets. We have not explored other combinatorial optimization methods (e.g., neural combinatorial optimization) due to their high computational cost.
>
> ```In response to the provided suggestions, we have conducted additional experiments with varying batch sizes, and the results are reported in Appendix C.5. These experiments show that the performance of the proposed ConBatch-BAL strategies relative to the baseline remains consistent across batch sizes.```
>
> >**W2:** The threshold is updated by subtracting the cost of the previously selected sample and dividing by the remaining steps. This is straightforward but may lead to inefficient sampling if a high-cost sample is selected early on, reducing the budget for remaining samples.
>
> By imposing budget thresholds, the dynamic thresholding-based strategy can mitigate the problem of selecting high-cost samples at initial batch steps. If the sample annotation cost is too high, the cost will exceed the imposed threshold, and the sample will not be selected, facilitating the collection of other points throughout the batch. However, this issue may be experienced by greedy approaches, as they can indeed select high-cost samples at the initial batch steps. We explain this aspect in Line 401.
>
> > Can the authors provide theoretical insights or guarantees about the convergence or stability of this threshold adjustment process could work? ...
>
> As detailed in Lines 188-190, acquiring samples based on mutual information satisfies submodularity (Kirsch et al., 2019). However, introducing cost-dependent samples imposes a knapsack constraint, which is NP-hard and may break submodularity, making it challenging to provide theoretical guarantees in this context.
>
> >**W3:** In real-world scenarios, annotation costs and uncertainty values can be highly variable....
>
> The threshold adjustment mechanism in dynamic thresholding is designed to redistribute the remaining budget dynamically. This adaptability helps mitigate the impact of extreme values, as the threshold recalibration at each step ensures that the budget is not overly influenced by any single outlier. However, the reviewer’s point can be still valid, as a limitation of the proposed dynamic thresholding strategy involves scenarios in which highly informative points cannot be selected because they are very costly, as explained in **W1**. We report this limitation in the paper and conduct a specific case study (i.e., cost_area) to quantify and explain the limitations (Lines 395-405).

---

> > ### Comment · Reviewer_7Kd8 · 2024-11-30
> >
> > I found the choice of dataset nieman17k would be more beneficial and useful for different batches size as nieman17k has more training data and the overall performance of nieman17k is worse than build_6k. It would be more convincing if the authors can show more datasets have better ablation results on dynamic thresholding. Meanwhile, as there is not more theoretical guarantees on dynamically choosing thresholding, I would lean towards keeping my score.

---

### Official Review · Reviewer_PBaM · 2024-11-04

**Soundness:** 3
**Presentation:** 3
**Contribution:** 2
**Rating:** 5
**Confidence:** 4

**Summary:**

This paper considers an active learning strategy with budget constraints. The proposed algorithm is based on dynamic thresholding and batch acquisition. Two real-world datasets are considered in experiments, with high cost and having geospatial constraints, under various budget and cost scenarios. The results show that the developed ConBatch-BAL strategies can reduce active learning iterations and data acquisition costs in real-world settings. The performance was compared concerning an infinite budget.  Therefore, the main study focuses on budge-constrained active learning with real-world two-image datasets.

**Strengths:**

The paper proposed an algorithm addressing the budget-constrained dynamic querying algorithm, which can be applied to the examined datasets. The analytic results show the trade-off between budget and informativeness (measures by mutual information). The appropriate strategy for reducing the labeling costs and gaining a large information under budget constraints is established and shows a potential to be applied in the real-world problem.

**Weaknesses:**

$\bullet$
The main issue is concerning $c(\bf x)$. It seems that there is no general $c(\bf x)$ to be applied for various datasets such as tablet datasets, conventional image datasets, and so on.  In the paper, only geospatial constraints are related to $c(\bf x)$. It weakens the contribution of the paper. When the image datasets can be considered, can the variance of each pixel be $c(\bf x)$? At least, the more general criteria for the $c(\bf x)$ can be required.

$\bullet$
Second, the range of experiments can be problematic on datasets. Combined with the first issue, there are few datasets to apply to the proposed algorithm. In active learning, there are datasets such as images, natural language, tablet datasets, and so on. The coverage of the proposed algorithm should not be all over the domains. However, the applicability or potential to other domains is important.

**Questions:**

$\bullet$
Can you consider any more metrics for $c(x),$ such as articular types of cost metrics that might be relevant for other real-world active learning scenarios beyond geospatial constraints?

$\bullet$
Can you provide any criteria or discussion for deciding max budget? Please let me know if you consider the specific factor for real datasets.

---

> ### Author Response · Authors · 2024-11-22
> **Response to Reviewer PBaM**
>
> Thank you for taking the time to review our paper. We have carefully addressed each of your comments below.
>
> > **W1:** The main issue is concerning c(x). It seems that there is no general c(x) to be applied for various datasets such as tablet datasets, conventional image datasets, and so on...
>
> Although the experiments showcased in the paper focus on label acquisition under geospatial constraints, **the problem statement formulated in Section 3.1 is general** and can accommodate other cost models, e.g., cost models and constraints defined in terms of monetary units or time.
>
> ```We have clarified the notation in the revised manuscript to clarify this aspect (Lines 186-188).```
>
> In our paper, we include a synthetic dataset derived from MNIST, where each digit is associated with a geolocation. In future works, we look forward to the creation of new synthetic datasets based on generic cost models (e.g., pixel-related metrics).
>
> >**W2:** Second, the range of experiments can be problematic on datasets. Combined with the first issue, there are few datasets to apply to the proposed algorithm...
>
> The proposed strategies cannot be directly tested on traditional active learning datasets, unless a synthetic cost model and relevant cost constraints are defined. Our work addresses the limitations of existing active learning strategies (and also existing datasets) by considering varying annotation costs and budget constraints in the (batch) selection process and releasing two relevant datasets. This need is born by real-world applications, e.g., predicting the energy performance of a city.
>
> We therefore introduce a budget-constrained problem formulation, release two geospatially constrained datasets, and propose two active learning strategies to tackle these challenges. This marks a first step toward applying active learning in practical scenarios, with opportunities for future research on diverse datasets, alternative cost models, and advanced optimization methods. A more detailed explanation can be found in the global response **G1**.
>
> ```In the revised manuscript, we encourage the development of novel real-world datasets with other cost model specifications (Lines 297-300).```
>
> >**Q1:** Can you consider any more metrics for c(x) such as articular types of cost metrics that might be relevant for other real-world active learning scenarios beyond geospatial constraints?
>
> Yes, other cost metrics other than distance can be defined following the general formulation introduced in the paper in Section 3.1.
>
> ```This has been clarified in the revised manuscript (Lines 185–188), where we explain that the cost model is problem-specific and can be defined according to the specific scenario.```
>
> >**Q2:** Can you provide any criteria or discussion for deciding max budget? Please let me know if you consider the specific factor for real datasets.
>
> In the conducted experiments, we consider two finite budgets (i.e., 2 km and 100 m) as well as a setting under an infinite budget. The 2-km budget is a sensible choice considering that an auditor will inspect a set of buildings in an area of the city, whereas the 100-m budget is an extreme case to verify the performance of the tested strategies under strict constraints.
>
> ```In the revised manuscript, we clarify that the budget is problem-specific and can be adapted for different real-world scenarios (Lines 185–188). ```

---

> > ### Comment · Reviewer_PBaM · 2024-11-26
> > **Response to Author**
> >
> > I appreciate the responses to my concerns. Though the proposal is valuable, the general use of this algorithm can be limited by the use of $c(\bx).$ Therefore, I keep my score.

---

### Author Response · Authors · 2024-11-22
**Global Response**

We thank all reviewers for their time and constructive feedback, which has greatly contributed to improving the quality of the manuscript. We are encouraged by your positive remarks regarding the strength of our contributions, which include (i) a new problem, (ii) two new real-world datasets to support it, and (iii) two algorithmic approaches to address it. We are also inspired by your critical remarks. With the provided responses and applied modifications, we hope to reinforce your support for our work. The revised paper has been submitted to OpenReview, with all modifications highlighted in blue for clarity.

In this global response, we address two comments raised by multiple reviewers:

**G1. Datasets tested.**
This work is motivated by the fact that existing active learning strategies do not consider varying label annotation costs or budget constraints within the (batch) selection process. In real-world applications, such as drone navigation, building energy audits, or infrastructure management planning, label annotation costs vary among candidate samples and budget constraints limit sample annotation. Based on this observation, we formulate the active learning problem under budget constraints (Eq. 2) and release two real-world datasets, in which label annotation is constrained by geospatial constraints.

To address the challenges that arise in the previously mentioned real-world application, we also introduce two active learning strategies for selecting samples for annotation under budget constraints. Keep in mind that we aim to enable the application of active learning methods to large-scale real-world scenarios (e.g. for estimating the energy performance of an entire city like Rotterdam). By releasing two new datasets, an active learning problem formulation, and two general algorithmic strategies therefor, we make a first step towards our goal.

Due to the above characteristics, this work definitely does not close a topic, but rather opens a new one. We look forward to further research involving new real-world datasets across diverse domains and cost models defined based on metrics beyond distance. We encourage the exploration of more advanced optimization approaches in future work.

Based on this discussion, we have included clarifications in the revised manuscript (Lines 185-188).

**G2. Impact of batch size on the proposed strategies.**

Following this comment, we have conducted additional experiments for varying batch sizes and reported the results in Appendix C.5. The experiments show that the performance of the proposed ConBatch-BAL strategies relative to the baseline remains consistent across batch sizes.

---

### Meta-Review · Area_Chair_xj5h · 2024-12-21

**Metareview:**

This paper addresses the problem of batch Bayesian active learning under budget constraints, introducing two heuristic strategies: dynamic thresholding and greedy acquisition. The proposed methods aim to balance annotation costs and uncertainty metrics for sample selection, with applications to real-world scenarios such as geolocated aerial imagery datasets. The authors also contribute two new datasets designed to test active learning strategies under budget constraints.

**Strengths**
* The paper introduces a novel problem formulation for batch Bayesian active learning under budget constraints.
* The inclusion of two new real-world datasets of geolocated aerial imagery adds value to the community.
* The proposed strategies (dynamic thresholding and greedy acquisition) are practically relevant for scenarios with varying annotation costs.

**Weaknesses**
* The cost function is defined for geospatial constraints, limiting applicability to other domains.
* The experiments are limited in scope, tested on only two real-world datasets, which causes generalizability concerns.
* The dynamic thresholding strategy lacks theoretical guarantees and shows sensitivity to hyperparameter choices.
* Comparisons are limited to a random baseline, omitting evaluations against other batch Bayesian active learning methods.

While the paper addresses a relevant and practical problem, the limitations in scope, theoretical grounding, and experimental evaluation outweigh its contributions. The authors are encouraged to broaden the generality of their cost function and provide stronger theoretical insights in future submissions.

**Additional Comments On Reviewer Discussion:**

The reviewers appreciated the paper’s focus on a novel problem and the introduction of two real-world datasets, but concerns about the generality of the cost function, limited experimental scope, lack of theoretical guarantees, and reliance on straightforward heuristics outweighed its strengths.

---

### Decision · Program_Chairs · 2025-01-22

Reject